# Is Your LLM Overcharging You? Tokenization, Transparency, and Incentives

Ander Artola Velasco [1]  Stratis Tsirtsis [2]  Nastaran Okati [3]  Manuel Gomez Rodriguez [1]

## Abstract

State-of-the-art large language models require specialized hardware and substantial energy to operate. Consequently, cloud-based services that provide access to these models have become very popular. In these services, the price users pay depends on the number of tokens a model uses to generate an output—they pay a fixed price per token. In this work, we show that this pricing mechanism creates a financial incentive for providers to strategize and misreport the (number of) tokens a model used to generate an output, and users cannot prove, or even know, whether a provider is overcharging them. However, we also show that, if an unfaithful provider is obliged to be transparent about the generative process used by the model, misreporting optimally without raising suspicion is hard. Nevertheless, as a proof-of-concept, we develop an efficient heuristic algorithm that allows providers to significantly overcharge users without raising suspicion. Crucially, the cost of running the algorithm is lower than the additional revenue from overcharging users, highlighting the vulnerability of users under the current pay-per-token pricing mechanism. Further, we show that, to eliminate the financial incentive to strategize, a pricing mechanism must price tokens linearly on their character count. While this makes a provider's profit margin vary across tokens, we introduce a simple prescription that allows a provider to maintain their average profit margin when transitioning to an incentive-compatible pricing mechanism. To complement our theoretical results, we conduct experiments with large language models from the `Llama`, `Gemma` and `Ministral` families, and prompts from a popular benchmarking platform.

[1]Max Planck Institute for Software Systems, Kaiserslautern, Germany [2]Hasso Plattner Institute, Potsdam, Germany [3]Max Planck Institute for Intelligent Systems, Tübingen, Germany. Correspondence to: Ander Artola Velasco <avelasco@mpi-sws.org>.

*Proceedings of the 43^{rd} International Conference on Machine Learning*, Seoul, South Korea. PMLR 306, 2026. Copyright 2026 by the author(s).

## 1. Introduction

Large language models (LLMs) are becoming ubiquitous across multiple industries—from powering chatbots and virtual assistants to driving innovation in research, healthcare, and finance (Romera-Paredes et al., 2024; Drori et al., 2022; Haupt & Marks, 2023; Li et al., 2023). However, since the computational resources required to run these models are significant, most (enterprise) users are unable to host them locally. As a result, users rely on a few cloud-based providers that offer LLMs-as-a-service to obtain access (Pais et al., 2022).

In a typical LLM-as-a-service, a user submits a prompt to the provider via an application programming interface (API). Then, the provider feeds the prompt into an LLM running on their own hardware, which (stochastically) generates a sequence of tokens as output using a generative process.[1] Finally, the provider shares the output with the user and charges them based on a simple pricing mechanism: a fixed price per token.[2] In this paper, we focus on the following fundamental question:

*What incentives does the pay-per-token pricing mechanism create for providers?*

Our key observation is that, in the interaction between a user and a provider, there is an asymmetry of information (Rasmusen, 1989; Mishra et al., 1998). The provider observes the entire generative process used by the model to generate an output, including its intermediate steps and the final output tokens, whereas the user only observes and pays for the (output) tokens shared with them by the provider. This asymmetry sets the stage for a situation known in economics as moral hazard (Holmström, 1979), where one party (the provider) has the opportunity to take actions that are not observable by the other party (the user) to maximize their own utility at the expense of the other party.

The core of the problem lies in the fact that the tokenization of a string is not unique. For example, consider that the user submits the prompt "What is the oldest

---

[1]Tokens are units that make up sentences and paragraphs, such as (sub-)words, symbols, and numbers.

[2]https://ai.google.dev/gemini-api/docs/pricing, https://openai.com/api/pricing.

city in the world?" to the provider, the provider feeds it into an LLM, and the model generates the output "|Dam|ascus|" consisting of two tokens. Since the user is oblivious to the generative process, a self-serving provider has the capacity to misreport the tokenization of the output to the user without even changing the underlying string. For instance, the provider could simply claim that the LLM generated the tokenization "|Da|ma|s|cus|" and overcharge the user for four tokens instead of two!

A simple remedy to build trust between the two parties would be to require providers to share with the user more information about the generative process used by the model, such as the next-token distribution in each step of the process. This would grant the user a form of (partial) auditability, since tokenizations, such as the one mentioned above, may have negligible probability in practice. Importantly, if the provider implements procedures to prevent the generation of low-probability tokens (*e.g.*, top-$p$ sampling (Holtzman et al., 2020), top-$k$ sampling), as commonly done in practice, such tokenizations would not only be unlikely, but rather implausible, giving grounds to the user to contest the specific tokenization of the output shared with them by the provider. In this case, a provider would have to invest additional effort (and resources) to misreport the tokenization of an output while preserving its plausibility, making such a strategic behavior significantly less worthy from a financial point of view.

However, some providers may be highly reluctant to share information that could potentially expose the internal workings of their LLMs, especially if the LLMs are proprietary and such information can be used by competitors (Carlini et al., 2024). In the absence of any additional means for the users to verify the truthfulness of the providers, the only remaining option is to regulate the transactions between users and providers in a way that eliminates the incentive for providers to engage in misreporting in the first place. To this end, we introduce and argue for a pay-per-character pricing mechanism that serves exactly this purpose.

**Our contributions.** We start by characterizing the problem of tokenization (mis-)reporting in LLMs as a principal-agent problem (Grossman & Hart, 1992; Dütting et al., 2024a). Building upon this characterization, we make the following contributions:

1. We show that, under pay-per-token, a provider's utility is tightly linked to the length of the reported output token sequence, creating an incentive for the provider to misreport an output's tokenization.

2. We show that, if the providers are transparent about the next-token distribution used by the LLMs they serve, they cannot expect to find the longest tokenization of an output that is plausible in polynomial time.

3. We show that, even under this form of transparency, the financial incentive for providers to (mis-)report tokenizations persists. As a proof-of-concept, we develop an efficient heuristic algorithm that allows providers to find plausible token sequences that are longer than or equal to a generated output token sequence, and is financially profitable—the additional revenue it yields is higher than the cost of running it.

4. We show that *any* incentive-compatible pricing mechanism must price tokens linearly on their character count. As a consequence, if each character is priced equally, there is only one incentive-compatible pricing mechanism, which we refer to as the pay-per-character pricing mechanism.

5. We show that, under an incentive-compatible pricing mechanism, a provider's profit margin varies across tokens. However, we introduce a simple prescription under which a provider who transitions from pay-per-token to pay-per-character can maintain their average profit margin.

Along the way, to illustrate and complement the above contributions, we conduct a series of experiments using LLMs from the Llama, Gemma and Ministral families and user input prompts from LMSYS Chatbot Arena, a popular platform for crowdsourced benchmarking (Zheng et al., 2024). Under the pay-per-token pricing mechanism, we empirically demonstrate that an unfaithful provider who is transparent about the generative process used by the LLM they serve can use our heuristic algorithm to overcharge users and, in many cases, obtain a significant financial gain. The code for our experiments is available at https://github.com/Human-Centric-Machine-Learning/token-pricing.

**Further related work.** Our work builds upon further related work on tokenization, economics of LLMs-as-a-service, mechanism design, and game theory in LLMs.

Multiple lines of empirical evidence have shown that tokenization plays a central role in developing and analyzing LLMs (Geh et al., 2024; Giulianelli et al., 2024; Geh et al., 2025; Petrov et al., 2023; Chatzi et al., 2025). Consequently, there have been numerous efforts to better understand and improve byte-pair encoding (BPE), the tokenization algorithm most commonly used in LLMs (Bostrom & Durrett, 2020; Zouhar et al., 2023; Lian et al., 2025; Sennrich et al., 2016). However, this line of work has overlooked the economic implications of tokenization (in the context of LLMs-as-a-service), which is the main focus of our work.

Within the rapidly growing literature on the economics of LLMs-as-a-service (La Malfa et al., 2024; Mahmood, 2024; Laufer et al., 2024; Cai et al., 2025; Saig et al., 2025; Chen

et al., 2026), most similar to ours is a very recent piece of work by Sun et al. (Sun et al., 2025), who study the problem of providers reporting artificially injected tokens during the hidden reasoning steps of recent LLMs. However, they limit their analysis to injecting additional reasoning tokens and do not consider alternative tokenization of the output string, which is the main focus of our work. In a related line of work, Cai et al. (Cai et al., 2025) and Saig et al. (Saig et al., 2025) also study a setting in which the provider has a financial incentive to be unfaithful to the users. However, in their setting, the provider has an incentive to be unfaithful about the LLM they use to generate outputs, rather than the tokenization of the outputs—it may use a cheaper-to-run LLM than the one it charges the users for. To reduce the financial incentive to strategize, Cai et al. (Cai et al., 2025) argue for solutions based on increased transparency as well as trusted execution environments, and Saig et al. (Saig et al., 2025) argue for a pay-for-performance pricing mechanism using a contract theory formulation.

The literature on mechanism design and game theory in LLMs has explored incentive auction mechanisms for generated content (Dütting et al., 2024b), LLM-augmented voting processes (Fish et al., 2024), and the potential of LLMs as economic agents (Filippas et al., 2024; Raman et al., 2024; Kovařík et al., 2023). However, to the best of our knowledge, our work is the first to explore incentive-compatible token pricing mechanisms in LLMs.

## 2. A Principal-Agent Model of Delegated Autoregressive Generation

We characterize the interaction between a user and an LLM provider as a principal-agent problem (Grossman & Hart, 1992), where the principal (the user) delegates a task (a generation) to the agent (the provider), who performs the task on behalf of the principal and gets paid based on a commonly agreed-upon contract (*i.e.*, a pricing mechanism). Although we consider the principal-agent framework as the most natural economic modeling approach for the interaction between a user and an LLM provider, it is worth highlighting that this particular type of principal-agent problem does not exactly match those that have been studied in standard contract theory (Bolton & Dewatripont, 2004; Dütting et al., 2024a). Therein, the principal is typically the one who designs the contract such that it incentivizes the agent to act in a way that aligns with the principal's interests. In contrast, in existing LLMs-as-a-service, the pricing mechanism is set unilaterally by the provider, leaving the user with (surprisingly) limited leverage.

In a typical interaction between a user and a provider under this characterization, the user first submits a prompt $q \in \Sigma^*$ to the provider, where $\Sigma^*$ denotes the set of all finite-length strings over an alphabet $\Sigma$. Then, the provider uses their own hardware to query an LLM with the prompt $q$, and the LLM (stochastically) generates an output token sequence $\mathbf{t} = (t_1, t_2, \ldots, t_k) \in \mathcal{V}^*$ in an autoregressive manner, one token at a time. Here, $t_i \in \mathcal{V}$ is the $i$-th token in a sequence of $k$ tokens, $\mathcal{V} \subset \Sigma^*$ is the (token) vocabulary used by the LLM,[3] and $\mathcal{V}^*$ denotes the set of all finite-length sequences over the vocabulary. Finally, the provider observes the generated sequence of tokens $\mathbf{t}$, and reports a token sequence $\tilde{\mathbf{t}} \sim \pi(\mathbf{t})$ to the user using a (non-deterministic) reporting policy $\pi$.

Before the interaction between a user and an LLM provider begins, both parties agree on a contract that specifies how the provider should be compensated for the output token sequence they report to the user. More specifically, the user and the provider agree on a *pricing mechanism* that determines the price $r(\tilde{\mathbf{t}})$ the user pays (*i.e.*, the revenue the provider earns) for the reported output token sequence $\tilde{\mathbf{t}}$:

**Definition 2.1** (Pricing mechanism). Given a vocabulary of tokens $\mathcal{V}$, a pricing mechanism is a function $r \colon \mathcal{V}^* \to \mathbb{R}_{\geq 0}$ that assigns a price to each reported output token sequence $\tilde{\mathbf{t}} \in \mathcal{V}^*$.

Throughout the paper, we focus on additive pricing mechanisms, which include the widely used pay-per-token pricing mechanism. An additive pricing mechanism independently assigns a price $r(\tilde{t}_i)$ to each token $\tilde{t}_i$ in a reported output token sequence $\tilde{\mathbf{t}}$, and calculates the price $r(\tilde{\mathbf{t}})$ of a reported output token sequence by adding up the price of each individual token.

Given a generated output token sequence $\mathbf{t}$ and a reported output token sequence $\tilde{\mathbf{t}} \sim \pi(\mathbf{t})$, the provider's utility $U_\pi(\tilde{\mathbf{t}}, \mathbf{t})$ is given by the difference between the revenue $r(\tilde{\mathbf{t}})$ they receive from the user for the reported sequence $\tilde{\mathbf{t}}$ and the energy costs $c_{\text{gen}}(\mathbf{t})$ and $c_\pi(\mathbf{t})$ of GPU computations[4] to generate the output sequence $\mathbf{t}$ and run the reporting policy $\pi$, respectively, *i.e.*,

$$U_\pi(\tilde{\mathbf{t}}, \mathbf{t}) = r(\tilde{\mathbf{t}}) - c_{\text{gen}}(\mathbf{t}) - c_\pi(\mathbf{t}). \tag{1}$$

In what follows, we refer to the reporting policy $\pi_0$ that always returns $\mathbf{t} = \pi_0(\mathbf{t})$ as the *truthful* policy, where note that $c_{\pi_0}(\mathbf{t}) = 0$ for all $\mathbf{t}$, and we denote the profit margin the provider obtains from generating and (truthfully) reporting the output token sequence $\mathbf{t}$ as $\rho(\mathbf{t}) = 1 - c_{\text{gen}}(\mathbf{t})/r(\mathbf{t}) = U_{\pi_0}(\mathbf{t}, \mathbf{t})/r(\mathbf{t})$.

Further, motivated by recent empirical studies showing that

---

[3]We assume $\Sigma \subset \mathcal{V}$ since this condition is satisfied by construction by most tokenization algorithms (Sennrich et al., 2016). Under this assumption, $\Sigma$ is the set of tokens that cannot be split further into other tokens and can include, for example, (non-)Latin characters.

[4]We focus on energy costs of GPU computations as they can be directly attributed to individual outputs. In practice, providers may have additional running costs, such as hardware maintenance, that they can amortize across outputs and incorporate into their pricing.

the energy an LLM consumes to generate an output $\mathbf{t}$ scales linearly with its length (Fernandez et al., 2025; Poddar et al., 2025), as well as our own experiments (refer to Figure 10 in Appendix D), we assume that $c_{\text{gen}}(\mathbf{t}) \approx c_o \cdot \text{len}(\mathbf{t})$, where $c_o \in \mathbb{R}_{>0}$ is a constant that represents the (energy) cost of generating a single token. Here, it is also worth clarifying that there are effective reporting policies that do not require GPU computations, that is, policies $\pi$ such that $c_\pi(\mathbf{t}) = 0$ for all $\mathbf{t}$, as those implemented by Algorithm 2. However, in Section 3, we will demonstrate that, in order not to raise suspicion, the reporting policies implemented by Algorithm 1 do require GPU computations.

Given a reported output token sequence $\tilde{\mathbf{t}}$, the user's utility $U_{\text{user}}(\tilde{\mathbf{t}})$ is given by the difference between the value $v(\tilde{\mathbf{t}})$ they derive from the sequence $\tilde{\mathbf{t}}$ and the price $r(\tilde{\mathbf{t}})$ they pay to the provider for $\tilde{\mathbf{t}}$, that is, $U_{\text{user}}(\tilde{\mathbf{t}}) = v(\tilde{\mathbf{t}}) - r(\tilde{\mathbf{t}})$. However, the user typically derives value from the text that the output token sequence represents, rather than the token sequence itself. For example, in creative writing, the user may be interested in the extent to which the generated text is captivating to read, and in code generation, the user may be interested in operational aspects of the generated code, such as its correctness and efficiency. Therefore, we assume that $v(\tilde{\mathbf{t}}) = v(\text{str}(\tilde{\mathbf{t}}))$, where $\text{str}: \mathcal{V}^* \to \Sigma^*$ maps a sequence of tokens to the respective string, and we use $|\text{str}(\tilde{\mathbf{t}})|$ to denote the number of characters in the string $\text{str}(\tilde{\mathbf{t}})$.

Since the user does not have direct access to the generated output sequence $\mathbf{t}$, a provider can, in principle, implement any reporting policy $\pi$ they prefer (*e.g.*, the one with the highest utility based on the pricing mechanism). However, arbitrary manipulations of the generated output may easily raise suspicion about the provider's practices. Therefore, motivated by the observation that LLMs can generate different token sequences corresponding to the same string (Geh et al., 2024; Cao & Rimell, 2021; Chirkova et al., 2023), we narrow our focus to reporting policies that misreport the tokenization of the generated output sequence *while preserving its string-level representation*. More formally, given a generated output token sequence $\mathbf{t}$ with $s = \text{str}(\mathbf{t})$, such reporting policies return token sequences $\tilde{\mathbf{t}}$ from the set $\mathcal{V}_s^* = \{\tilde{\mathbf{t}} \in \mathcal{V}^* : \text{str}(\tilde{\mathbf{t}}) = s\}$. Then, it is easy to see that, as long as there exists a token sequence $\tilde{\mathbf{t}} \in \mathcal{V}_s^*$ such that $r(\tilde{\mathbf{t}}) > r(\mathbf{t})$, a reporting policy $\pi$ that (mis-)reports $\mathbf{t}$ as $\tilde{\mathbf{t}}$ may achieve higher utility than the reporting policy $\pi_0$ that truthfully reports $\mathbf{t}$, *i.e.*,

$$U_\pi(\tilde{\mathbf{t}}, \mathbf{t}) > U_{\pi_0}(\mathbf{t}, \mathbf{t}), \quad \text{and} \quad v(\tilde{\mathbf{t}}) = v(\mathbf{t}).$$

In other words, the provider has an incentive not to be truthful and potentially overcharge the user, and they can do so in a way that maintains the value the user derives from the reported output sequence. In what follows, we will explore the conditions under which such strategic behavior can occur and remain undetected by the user. Later on, we will propose a pay-per-character pricing mechanism that provably eliminates the provider's incentive for this type of strategic behavior.

## 3. Provider Incentives Under the Pay-Per-Token Pricing Mechanism

In this section, we analyze the pay-per-token pricing mechanism using the principal-agent model introduced in Section 2. First, we show that, under this mechanism, the provider's utility is tightly linked to the length of the reported output token sequence—the longer the reported sequence, the higher the provider's utility—and this creates an incentive for the provider to (mis-)report an output's tokenization. Then, we show that requiring the provider to be transparent about the next-token distributions used by the LLM they serve places obstacles in their ability to misreport tokenizations, as they cannot expect to find the longest plausible tokenization of a given output in polynomial time. However, we demonstrate that, in practice, this computational hardness does not preclude the provider from efficiently finding plausible tokenizations of a given output that increase their utility.

### 3.1. Pay-Per-Token Incentivizes (Mis-)Reporting Longer Tokenizations

To be profitable, a cloud-based LLM provider needs to at least cover their costs of generating outputs. Therefore, under the assumption that the cost of generating a single output is a linear function of its length, the widely used pay-per-token pricing mechanism is a natural choice.

**Definition 3.1** (Pay-per-token)**.** A pricing mechanism $r: \mathcal{V}^* \to \mathbb{R}_{\geq 0}$ is called pay-per-token if and only if it is additive and, for all $t \in \mathcal{V}$, it satisfies $r(t) = r_o$, where $r_o \geq 0$ is a constant price per token.

As an immediate consequence, under the pay-per-token pricing mechanism, the revenue that the provider earns from (mis-)reporting an output token sequence $\tilde{\mathbf{t}}$ is a linear function of the output length, *i.e.*, $r(\tilde{\mathbf{t}}) = r_o \cdot \text{len}(\tilde{\mathbf{t}})$. Further, since the cost to generate the output sequence $\mathbf{t}$ is independent of the reported output sequence $\tilde{\mathbf{t}}$, it is easy to see that, for any pair of reporting policies $\pi$ and $\pi'$ with the same running cost $c_\pi(\mathbf{t}) = c_{\pi'}(\mathbf{t})$, and for all $\tilde{\mathbf{t}} \sim \pi(\mathbf{t})$ and $\tilde{\mathbf{t}}' \sim \pi'(\mathbf{t})$ such that $\text{len}(\tilde{\mathbf{t}}) > \text{len}(\tilde{\mathbf{t}}')$, it holds that

$$U_\pi(\tilde{\mathbf{t}}, \mathbf{t}) > U_{\pi'}(\tilde{\mathbf{t}}', \mathbf{t}). \tag{2}$$

Therefore, a rational provider has a financial incentive to use reporting policies $\pi$ favoring tokenizations longer than those generated by the LLM they serve, *i.e.*, $\text{len}(\tilde{\mathbf{t}}) > \text{len}(\mathbf{t})$

for any generated output $\mathbf{t}$ and $\tilde{\mathbf{t}} \sim \pi(\mathbf{t})$. Crucially, this incentive can be further amplified in the presence of competition among providers. To see this, consider a simple scenario in which the market consists of a single (faithful) provider charging users a price per token $r_o$. Now, suppose that a second (unfaithful) provider enters the market, serves the same LLM, and charges a lower price per token $r'_o = \alpha \cdot r_o$ for some $\alpha \in (0, 1)$, possibly even below the cost $c_o$ of generating it (Williamson, 1977). Using a reporting policy $\pi$ that, for each (sufficiently long) generated output sequence $\mathbf{t}$, misreports a token sequence $\tilde{\mathbf{t}}$ with $\texttt{len}(\tilde{\mathbf{t}}) \approx \frac{1}{\alpha} \cdot \texttt{len}(\mathbf{t})$, the unfaithful provider can effectively earn the same revenue per generated output sequence as their competitor, *i.e.*, $r'_o \cdot \texttt{len}(\tilde{\mathbf{t}}) \approx r_o \cdot \texttt{len}(\mathbf{t})$. At the same time, due to advertising a lower price per token, they can earn a strictly higher total revenue than their competitor by attracting a larger portion of the user base, making misreporting tokenizations an effective (but unfair) strategy to dominate the market.

A natural choice among reporting policies that increase the length of reported tokenizations is the policy $\pi^{\mathrm{R}}_m$, which constructs $\tilde{\mathbf{t}}$ by (randomly) splitting up to $m$ tokens in $\mathbf{t}$ and, as shown in Algorithm 2 in Appendix A, can be implemented with no GPU computations. Clearly, the financial incentive for (mis-)reporting tokenizations using $\pi^{\mathrm{R}}_m$ increases (linearly) with $m$, until all reported tokens become single characters, as shown in Figure 1a. However, the larger the value of $m$, the lower the chances that the reported tokenization $\tilde{\mathbf{t}}$ could have been generated by the LLM the provider serves, as shown in Figure 1b. This observation lends support to the idea that the provider should not be required to report only an output sequence, but also the next-token probability corresponding to each token in the sequence, offering the user the means to contest a reported output token sequence.[5] Next, we will show that an unfaithful provider who aims to find the reporting policy $\pi$ with the highest utility among those under which $\tilde{\mathbf{t}} \sim \pi(\mathbf{t})$ is always plausible for any generated output $\mathbf{t}$ is likely to fail.

### 3.2. Misreporting Optimally Without Raising Suspicion Is Hard

In this section, we will focus on a setting in which the provider is transparent about next-token probabilities and implements top-$p$ sampling (Holtzman et al., 2020), a widely used sampling technique that, given a (partial) token sequence $\mathbf{t}$, restricts the sampling of the next token to the smallest set $\mathcal{V}_p(\mathbf{t}) \subseteq \mathcal{V}$ whose cumulative next-token probability is at least $p \in (0, 1)$. Then, given a generated output sequence $\mathbf{t}$ with $s = \texttt{str}(\mathbf{t})$, the provider aims to find a reporting policy $\pi$ with the highest utility among those un-

---

[5]Users with access to large amounts of reported output token sequences may be able to contest them using statistical analysis techniques (see Section 5).

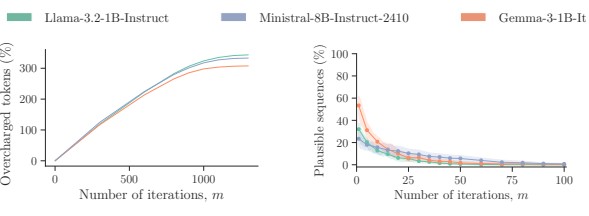

*(a)* Overcharged tokens     *(b)* Plausible tokenizations

*Figure 1.* **Misreporting the tokenization of LLM outputs using the policies $\pi^{\mathrm{R}}_m$.** Panel (a) shows the percentage of tokens overcharged by an unfaithful provider who misreports the tokenization of the outputs to 600 prompts from the LMSYS Chatbot Arena platform using the policies $\pi^{\mathrm{R}}_m$, for different values of $m$. Panel (b) shows the fraction of outputs where Algorithm 2 returns a tokenization that is plausible under top-$p$ sampling. In both panels, we set the temperature to 1.3 and use top-$p$ sampling with $p = 0.95$.

der which $\tilde{\mathbf{t}} \sim \pi(\mathbf{t})$ is always plausible. In what follows, we will show that the provider cannot expect to find such a policy in polynomial time.

Our starting point is the realization that, under the pay-per-token pricing mechanism, the problem of finding a policy $\pi$ with the highest utility among those under which $\tilde{\mathbf{t}} \sim \pi(\mathbf{t})$ is always plausible is at least as hard as finding the longest plausible tokenization $\tilde{\mathbf{t}}$ of $s$, *i.e.*,

$$\max_{\tilde{\mathbf{t}} \in \mathcal{V}^*_s} \quad \texttt{len}\left(\tilde{\mathbf{t}}\right)$$
$$\text{subject to} \quad \tilde{t}_i \in \mathcal{V}_p(\tilde{\mathbf{t}}_{\leq i-1}) \; \forall i \in [\texttt{len}(\tilde{\mathbf{t}})], \tag{3}$$

where $\tilde{\mathbf{t}}_{\leq i-1} = (\tilde{t}_1, \ldots, \tilde{t}_{i-1})$ is the prefix of the reported output sequence up to the $i$-th token. This is because this latter problem is a particular instance of the problem the provider aims to solve, where the cost $c_\pi(\mathbf{t})$ is constant for all output token sequences $\mathbf{t}$ and all candidate policies $\pi$ available to the provider. In this context, it is worth noting that a provider may be interested in solving this particular instance of the problem if, for example, they are considering only policies that check (once) the plausibility of a modified sequence $\tilde{\mathbf{t}} \neq \mathbf{t}$, such as the policy implemented by Algorithm 1 in Section 3.3.

The following theorem, whose proof is given in Appendix C, tells us that, in general, one cannot expect to find the longest plausible tokenization $\tilde{\mathbf{t}}$ of $s$ in polynomial time and, hence, a provider aiming to find a policy $\pi$ with the highest utility among those under which $\tilde{\mathbf{t}} \sim \pi(\mathbf{t})$ is always plausible is likely to fail:

**Theorem 3.2.** *The problem of finding the longest tokenization of a given output that is plausible under top-$p$ sampling, as defined in Eq. 3, is NP-Hard.*

The proof of the above theorem relies on a reduction from the Hamiltonian path problem (Karp, 2010). More specifically, given a graph, it creates an instance of our problem

---
**Algorithm 1** It returns a longer plausible token sequence

---
**Input** True output token sequence $\mathbf{t}$, number of iterations $m$, token-to-id function $\mathtt{id}(\bullet)$

**Initialize** $\hat{\mathbf{t}} \leftarrow \mathbf{t}$

**for** $m$ iterations **do**

$\quad i \leftarrow \operatorname{argmax}_{j \in [\mathtt{len}(\hat{\mathbf{t}})]} \mathtt{id}(\hat{t}_j)$

$\quad$ **if** $|\mathtt{str}\left(\hat{t}_i\right)| = 1$ **then break**

$\quad (t'_1, t'_2) \leftarrow \underset{v_1, v_2 \in \mathcal{V}}{\operatorname{argmax}} \left\{ \min\left(\mathtt{id}(v_1), \mathtt{id}(v_2)\right) \right.$

$\qquad\qquad\qquad \left. : \mathtt{str}\left((v_1, v_2)\right) = \mathtt{str}\left(\hat{t}_i\right) \right\}$

$\quad \hat{\mathbf{t}} \leftarrow (\hat{\mathbf{t}}_{<i}, t'_1, t'_2, \hat{\mathbf{t}}_{>i})$

**end for**

**if** $\mathtt{plausible}(\hat{\mathbf{t}})$ **then** $\tilde{\mathbf{t}} \leftarrow \hat{\mathbf{t}}$ **else** $\tilde{\mathbf{t}} \leftarrow \mathbf{t}$

**Return** $\tilde{\mathbf{t}}$

---

that establishes a one-to-one correspondence between a path that does not visit any node twice and a token sequence that is plausible only if it does not include any token twice. In Appendix C.1.1, we show that the above hardness result can be extended to a setting in which the provider implements top-$k$ sampling and, in Appendix C.1.2, we show that it can also be extended to a setting in which the provider does not implement any procedure to prevent the generation of low-probability tokens but aims to report sequences whose generation probability is greater than a given threshold.

Further, the above hardness result readily implies that there exists a computational barrier that precludes an unfaithful transparent provider from optimally benefiting from misreporting without raising suspicion. However, we will next demonstrate that, in practice, it does not rule out the possibility that a provider efficiently finds a reporting policy $\pi$ under which $\tilde{\mathbf{t}} \sim \pi(\mathbf{t})$ is always plausible for any generated output $\mathbf{t}$ offering higher utility than the (truthful) reporting policy $\pi_0$.

### 3.3. Can a Provider Overcharge a User Without Raising Suspicion?

We answer this question affirmatively. As a proof-of-concept, we introduce a simple heuristic algorithm that, given a generated output sequence $\mathbf{t}$ with $s = \mathtt{str}(\mathbf{t})$, finds a plausible tokenization $\tilde{\mathbf{t}}$ of $s$ longer than or equal to $\mathbf{t}$, and show that this algorithm can be used to construct a reporting policy $\pi_m^{\text{H}}$ offering higher utility than the (truthful) reporting policy $\pi_0$. Here, our goal is to demonstrate that, under the pay-per-token pricing mechanism predominantly used by cloud providers of LLM-as-a-service, users are indeed vulnerable to self-serving providers who may overcharge them without raising suspicion.

Our heuristic algorithm, summarized in Algorithm 1, is based on the key empirical observation that, given the most likely tokenization $\mathbf{t}$ of a string $s = \mathtt{str}(\mathbf{t})$, alternative tokenizations of $s$ that are not *too different* from $\mathbf{t}$ are very likely to be plausible, as shown in Figure 1b. In a nutshell, our algorithm starts from a given output sequence $\mathbf{t}$ and iteratively splits tokens in it for a number of iterations $m$ specified by the provider. In each iteration, the algorithm selects the token with the highest index in the vocabulary and, if it is longer than one character, it splits it into a pair of new tokens with the highest minimum index in the vocabulary whose concatenation maps to the same string.[6] The algorithm continues either until it has performed $m$ splits or all tokens in the sequence are single characters, in which case it terminates the loop. Finally, it checks whether the resulting token sequence $\hat{\mathbf{t}}$ is plausible and, if it is indeed plausible, it reports it to the user. For example, under top-$p$ sampling, evaluating plausibility reduces to checking whether $\hat{t}_i \in \mathcal{V}_p(\hat{\mathbf{t}}_{\leq i-1})$ for all $i \in [\mathtt{len}(\hat{\mathbf{t}})]$. However, our algorithm is agnostic to the choice of plausibility criteria (refer to Appendices C.1.1 and C.1.2 for alternatives). If $\hat{\mathbf{t}}$ is not plausible, the algorithm reports the true output token sequence $\mathbf{t}$.

Using prompts from the LMSYS Chatbot Arena platform, we find empirical evidence that, despite its simplicity, the reporting policy $\pi_m^{\text{H}}$ implemented by Algorithm 1 succeeds at helping a provider overcharge users whenever they serve LLMs with temperature values $>1.0$, such as those commonly used in creative writing tasks. Figure 2 summarizes the results, which show that, for example, for `Llama-3.2-1B-Instruct`, a provider who uses Algorithm 1 can overcharge users by up to $11.2\%$, $1.8\%$ and $0.28\%$, respectively for $p = 0.99, 0.95, 0.90$. Moreover, the results also reveal that the additional revenue is unimodal with respect to the number of iterations $m$, and the optimal value of $m$ decreases as $p$ decreases and achieving plausibility becomes harder. This is because, for large values of $m$, the token sequence $\hat{\mathbf{t}}$ resulting from iteratively splitting tokens becomes less likely to be plausible, as shown in Figure 4 in Appendix D.1. However, if plausible, it does provide a strictly larger additional revenue.

The above experimental results show that Algorithm 1 allows a provider to obtain additional revenue when exclusively misreporting plausible tokenizations. However, they do not directly imply that the provider's utility would increase. This is because, in order to verify the plausibility of $\hat{\mathbf{t}}$, the reporting policy $\pi_m^{\text{H}}$ implemented by Algorithm 1 requires the evaluation of the next-token probabilities that a model assigns to $\hat{\mathbf{t}}$ using a forward pass, *i.e.*, $c_{\pi_m^{\text{H}}}(\mathbf{t}) > 0$ in Eq. 1. In what follows, we will demonstrate that the additional revenue from misreporting can largely surpass the cost of verifying plausibility, making misreporting a

---
[6]We focus on splitting tokens based on their index, motivated by the BPE algorithm, where tokens with higher indices are (generally) longer, and hence are more likely to result in a plausible tokenization. Refer to Appendix D.2 for concrete examples.

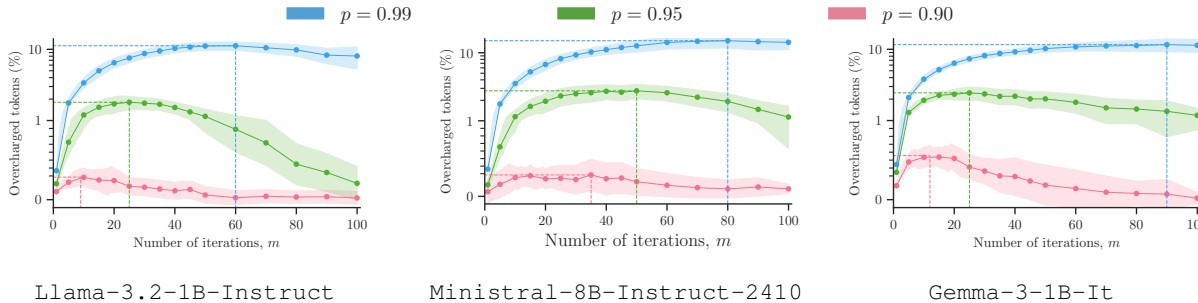

*Figure 2.* **Additional revenue from misreporting the tokenization of outputs using Algorithm 1.** The panels show the percentage of tokens overcharged by an unfaithful provider who misreports the tokenization of the outputs generated by an LLM to 600 prompts from the LMSYS Chatbot Arena platform using the reporting policies $\pi_m^{\text{H}}$ implemented by Algorithm 1, for different values of $m$ and $p$. The dashed vertical lines correspond to the optimal value of $m$. Here, we set the temperature of the model to 1.3. Refer to Appendix D.1 for additional results using alternative temperature values and other LLMs.

financially profitable strategy.

The key insight is that, as we demonstrate empirically in Figure 10 and Table 1 of Appendix D, and in agreement with recent works (Fernandez et al., 2025), the energy costs of GPU computations required to evaluate the probability a model assigns to a tokenization does not depend on the specific sequence (or its length). Therefore, we can readily conclude that $c_{\pi_m^{\text{H}}}(\mathbf{t}) = c_{\text{v}}$ is constant and, for a given number of iterations $m$, the reporting policy $\pi_m^{\text{H}}$ implemented by Algorithm 1 offers a gain in average utility with respect to the (truthful) policy $\pi_0$ if and only if

$$\mathbb{E}\left[\texttt{plausible}\left(\hat{\mathbf{t}}\right)\right] \cdot m \cdot r_o > c_{\text{v}}$$
$$\iff \rho(\mathbf{t}) > 1 - \mathbb{E}\left[\texttt{plausible}\left(\hat{\mathbf{t}}\right)\right] \cdot m \cdot \frac{c_o}{c_{\text{v}}}, \quad (4)$$

where the expectation over $\hat{\mathbf{t}}$ implicitly depends on the choice of $m$ and $\rho(\mathbf{t}) = 1 - c_{\text{gen}}(\mathbf{t})/r(\mathbf{t}) = 1 - c_o/r_o = \rho_o$ is the provider's profit margin under a pay-per-token pricing mechanism, which is constant since both $c_{\text{gen}}(\mathbf{t})$ and $r(\mathbf{t})$ are directly proportional to $\texttt{len}(\mathbf{t})$.

Using the same prompts from the LMSYS Chatbot Arena platform as above and the optimal number of iterations $m$ shown in Figure 2, we find that, for a wide range of the provider's profit margin values $\rho_o$, the reporting policy $\pi_m^{\text{H}}$ implemented by Algorithm 1 offers higher average utility than the (truthful) policy $\pi_0$. Figure 3 summarizes the results, which show that, for example, for Llama-3.2-1B-Instruct, the utility gain surpasses 10.5% for $p = 0.99$ regardless of the margin, and reaches 1.7% and 0.4% for $p = 0.95$ and $p = 0.90$, respectively.

The above empirical results demonstrate that, even if a provider is transparent about next-token probabilities and (mis-)reports only plausible tokenizations, users remain vulnerable to the (potentially) unfaithful behavior from the provider. To address this vulnerability, in the next section, we study incentive-compatible pricing mechanisms

that provably eliminate the provider's incentive to misreport an output token sequence.

## 4. Incentive-Compatible Pricing Mechanisms

To eliminate the provider's incentive to misreport an output token sequence, in this section, we look into the design of incentive-compatible pricing mechanisms. Incentive-compatibility is a (desirable) property studied in mechanism design (Nisan & Ronen, 2001) that, in the context of our work, ensures that the pricing mechanism creates no economic incentive for the provider to misreport an output token sequence—they cannot benefit from not telling the truth.

**Definition 4.1.** A pricing mechanism $r$ is incentive-compatible if and only if, for any generated output token sequence $\mathbf{t} \in \mathcal{V}^*$, any reporting policy $\pi$ and any $\tilde{\mathbf{t}} \sim \pi(\mathbf{t})$, it holds that $U_{\pi_0}(\mathbf{t}, \mathbf{t}) \geq U_\pi(\tilde{\mathbf{t}}, \mathbf{t})$, where $\pi_0$ is the policy that faithfully reports the generated sequence, *i.e.*, $\pi_0(\mathbf{t}) = \mathbf{t}$.

Importantly, if a pricing mechanism satisfies incentive-compatibility, the revenue a provider earns for reporting an output token sequence $\tilde{\mathbf{t}}$ depends only on the string $s = \texttt{str}\left(\tilde{\mathbf{t}}\right)$ and not on the token sequence itself, as shown by the following proposition:

**Proposition 4.2.** *If a pricing mechanism $r$ is incentive-compatible, then, for all $\tilde{\mathbf{t}}, \tilde{\mathbf{t}}' \in \mathcal{V}^*$ such that $\texttt{str}\left(\tilde{\mathbf{t}}\right) = \texttt{str}\left(\tilde{\mathbf{t}}'\right)$, it holds that $r\left(\tilde{\mathbf{t}}\right) = r\left(\tilde{\mathbf{t}}'\right)$.*

Perhaps surprisingly, the above proposition readily allows us to provide a simple characterization of the family of incentive-compatible pricing mechanisms. In particular, the following theorem tells us that it consists of all mechanisms that charge for an output sequence $\mathbf{t}$ linearly on its character counts:

**Theorem 4.3.** *A pricing mechanism $r$ is additive and*

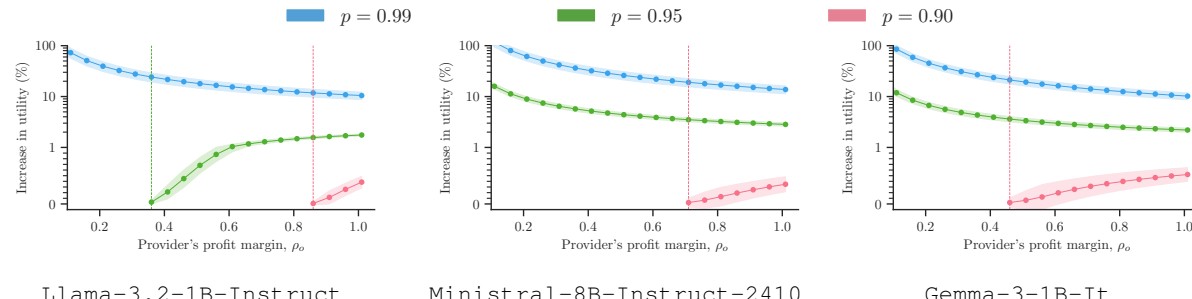

Llama-3.2-1B-Instruct      Ministral-8B-Instruct-2410      Gemma-3-1B-It

*Figure 3.* **Financial gain from misreporting the tokenization of outputs using Algorithm 1.** The panels show the utility gain that an unfaithful provider who misreports the tokenization of the outputs generated by an LLM to 600 prompts from the LMSYS Chatbot Arena platform using Algorithm 1 can achieve, for different values of $p$ and the provider's profit margin $\rho_o$. The dashed vertical lines represent the minimum margin above which misreporting is financially viable according to Eq. 4. Here, for each value of $p$, we run Algorithm 1 using the optimal number of iterations $m$ shown in Figure 2, set the temperature of the model to 1.3. Refer to Figure 6 of Appendix D.1 for additional results using alternative temperature values and other LLMs.

*incentive-compatible if and only if*

$$r(\mathbf{t}) = \sum_{\sigma \in \Sigma} \mathtt{count}_\sigma(\mathbf{t}) \cdot r(\sigma) \text{ for all } \mathbf{t} \in \mathcal{V}^*, \quad (5)$$

*where* $\mathtt{count}_\sigma(\mathbf{t})$ *counts the number of occurrences of the character* $\sigma$ *in* $\mathtt{str}(\mathbf{t})$.

This characterization directly implies that, if the provider decides to assign the same price $r_c$ to each character $\sigma \in \Sigma$, there exists only one incentive-compatible pricing mechanism, *i.e.*, $r(\mathbf{t}) = |\mathtt{str}(\mathbf{t})| \cdot r_c$, which we refer to as the *pay-per-character* pricing mechanism. Moreover, the above theorem also implies that, as long as the vocabulary $\mathcal{V}$ contains tokens composed of more than one character, which is typically the case, the pay-per-token pricing mechanism is provably not incentive-compatible.

Further, it is easy to see that the revenue $r(\mathbf{t})$ earned by a provider who adopts an incentive-compatible pricing mechanism cannot be directly proportional to the energy costs $c_{\text{gen}}(\mathbf{t}) \approx c_o \cdot \mathtt{len}(\mathbf{t})$ of GPU computations required to generate the output sequence $\mathbf{t}$. Consequently, the provider's profit margin $\rho(\mathbf{t}) = 1 - c_{\text{gen}}(\mathbf{t})/r(\mathbf{t})$ fluctuates with the number of characters per token in the output sequence $\mathbf{t}$: higher values yield higher profit margins, while lower values yield lower profit margins. This is because, given a fixed string $s$, the provider's revenue is independent of the output's tokenization due to Proposition 4.2 and their profit margin increases as the length of $\mathbf{t}$ (and the energy costs associated with generating it) decreases.

At the same time, it also introduces an operational challenge for providers: under incentive-compatible pricing, a provider's profit margin cannot be determined exactly in advance, as it depends on the user prompt and the number of characters in the respective output sequence. However, as long as the number of characters in an output sequence positively correlates with the number of tokens—

which is arguably the case in most realistic scenarios—the revenue and energy costs of generation do remain proportional on expectation over the distribution of (outputs to) user prompts under a pay-per-character pricing mechanism.

Building on this observation, a simple prescription that allows a faithful provider to transition from pay-per-token to pay-per-character pricing while preserving their *average* profit margin is to set the per-character price $r_c = r_o \cdot \mathtt{tpc}$, where $\mathtt{tpc}$ is the (empirical) average ratio of tokens to characters.

Using responses to prompts from the LMSYS Chatbot Arena platform, we find that a provider who transitions from pay-per-token to pay-per-character following the above prescription maintains their average profit margin, as expected. Moreover, as shown in Figure 11 in Appendix D.4, the provider's profit margin under pay-per-character is positive for the (vast) majority of output sequences, and it is only negative for a small number of output sequences. For example, for Llama-3.2-1B-Instruct, the provider's profit margin under pay-per-character is positive for 92.4%, 93.3%, 94.1% of the output sequences for $\rho_o = 0.2, 0.4$, and 0.6, respectively.

## 5. Discussion and Limitations

In this section, we discuss several assumptions and limitations of our work and propose avenues for future work.

**Model assumptions.** We have focused on additive pricing mechanisms, which include the widely used pay-per-token mechanism. It would be interesting to analyze provider incentives under other families of pricing mechanisms proposed in the literature, such as those based on the quality of the generated text (Saig et al., 2025). Further, we note that, while the hardness result in Theorem 3.2 holds without additional assumptions, providers would arguably only be forced

to report plausible tokenizations if the users have a mechanism to verify that a token sequence is in fact plausible. This raises the question of rigorously defining the information that both parties have access to when interacting, and how to verify that tokens are generated according to the advertised inference process. We leave this for future work and point towards trusted execution environments (Jauernig et al., 2020) and zero-knowledge proofs (Sun et al., 2024) as potential solutions to this problem.

**Methods.** To demonstrate the vulnerability of users under the pay-per-token pricing mechanism, we have introduced a heuristic algorithm that allows the provider to increase their utility by finding longer yet plausible tokenizations of the true output token sequence. However, there may exist other, more sophisticated methods for the provider to take advantage of the pay-per-token pricing mechanism, and there may also exist ways to defend users against such malicious behavior, other than a change of the pricing mechanism (Chatzi et al., 2026). In this context, it would also be interesting to develop statistical methods to audit the tokenizations reported by a provider (Velasco et al., 2026). Further, it is important to note that providers of proprietary models can also misreport other aspects of the generative process—such as the next-token distributions—or publicly release a tokenizer different from the one used internally by the model, leaving users with no means of detecting whether a provider is overcharging them. Lastly, we note that, similarly to pay-per-token, the pay-per-character pricing does not prevent the provider from artificially increasing the number of characters in the output string by increasing the model's verbosity.

**Evaluation.** We have conducted experiments with state-of-the-art open-weight LLMs using different tokenizers and architectures. It would be interesting to conduct experiments with proprietary LLMs, which are widely used in practice; however, we cannot find any reason to believe that our results will not generalize to proprietary LLMs. Further, we have illustrated our theoretical results using prompts from the LMSYS Chatbot Arena platform, which, despite being the most widely used for LLM evaluation based on pairwise comparisons, has been recently criticized (Singh et al., 2026; Zhou et al., 2023) and may not always reflect the real-world distribution of user prompts.

Further, we would like to emphasize that our work deliberately focuses on the micro level (Mas-Colell et al., 1995), modeling incentives in the isolated interaction between a single user and a single provider. However, altering the pricing mechanism could generate non-trivial downstream effects across the broader LLM-as-a-service ecosystem—influencing how users select providers and the resulting competitive landscape—and extending the analysis to these broader market dynamics remains an interesting direction for future work.

**Legal, reputational, and regulatory considerations.** One could argue that, despite the lack of incentive-compatibility of the current pay-per-token pricing mechanism, token misreporting is unlikely to occur as it would require the provider to violate their terms of service and risk damaging their reputation. However, history has shown that whenever economic incentives are strong, contracts are frequently breached and reputation is put at risk. In this context, a very prominent, relevant example is the price fixing scandal involving Google's second-price auctions (U.S. Department of Justice, Office of Public Affairs, 2025). Following this scandal, Google was pushed to shift from second-price to first-price auctions.

More broadly, in markets for credence goods, *i.e.*, services where quality or quantity is difficult for buyers to verify (Balafoutas & Kerschbamer, 2020), even highly regulated entities have systematically exploited information asymmetries in their own self-interest: the automotive industry has falsified environmental emissions data (Bachmann et al., 2019), the tobacco industry has manipulated scientific studies and advertising for decades (Bero, 2005), and the financial sector has engaged in market manipulation (Hou & Skeie, 2014; Markham, 2015; Tayan, 2019; Noel & Osman, 2024). In the current market of LLMs-as-a-service—which increasingly consists of a long tail of smaller providers who have less reputation to lose—we show that a systematic vulnerability exists, and that the economic incentives to exploit it are significant. With legislation and regulatory mechanisms still trying to catch up with advancements in generative AI,[7] we argue that user protection is best ensured by calling for pricing mechanisms with incentive compatibility guarantees.

# 6. Conclusions

In this work, we have studied the financial incentives of cloud-based providers in LLM-as-a-service using a principal-agent model of delegated autoregressive generation. We have shown that pay-per-token inadvertently creates a financial incentive for providers to misreport the number of tokens in the response of an LLM. To address this vulnerability, we proved that providers must be required to price users per character rather than per token. More broadly, we hope that our findings will encourage further scrutiny of the incentives shaping current practices in the LLM-as-a-service market.

---

[7]Legislation such as the EU AI Act (regulation 2024/1689 of the European Parliament) is primarily aimed at mitigating safety and fundamental rights risks, rather than verifying billing accuracy.

## Acknowledgements

Gomez-Rodriguez acknowledges support from the European Research Council (ERC) under the European Union's Horizon 2020 research and innovation programme (grant agreement No. 945719 and 101169607). Tsirtsis acknowledges supports from the Alexander von Humboldt Foundation in the framework of the Alexander von Humboldt Professorship (Humboldt Professor of Technology and Regulation awarded to Sandra Wachter) endowed by the Federal Ministry of Education and Research via the Hasso Plattner Institute.

## Impact Statement

As large language models (LLMs) become increasingly integrated into the 21st-century economy, millions of users rely on a market of cloud-based services for access. Our work sheds light on the economic incentives that the pay-per-token pricing mechanism inadvertently creates. On the positive side, we believe that our work can spark a discussion on the need for more transparent and fair pricing mechanisms in the LLM ecosystem. On the flip side, the heuristic algorithm we introduce could be misused by an unfaithful provider to overcharge users. However, we emphasize that our intention is to use it as a proof-of-concept, and not as an algorithm to be deployed in practice, similarly to the broader literature on adversarial attacks in machine learning (Szegedy et al., 2014; Goodfellow et al., 2015; Chakraborty et al., 2021).

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

# A. Misreporting policies $\pi_m^{\text{R}}$ with random token splitting

Here, we present the implementation of the policies $\pi_m^{\text{R}}$, which construct a longer tokenization $\tilde{\mathbf{t}}$ by randomly splitting tokens in the original tokenization $\mathbf{t}$.

---

**Algorithm 2** It returns a token sequence $\tilde{\mathbf{t}}$ longer than or equal to $\mathbf{t}$

---

    **Input** Generated output token sequence $\mathbf{t}$, number of iterations $m$
    **Initialize** $\tilde{\mathbf{t}} \leftarrow \mathbf{t}$
    **for** $m$ iterations **do**
        $\texttt{valid\_splits} \leftarrow \{(i, t_1', t_2') \text{ such that } i \in [\texttt{len}(\tilde{\mathbf{t}})], t_1', t_2' \in \mathcal{V}, \text{ and } \texttt{str}(t_1', t_2') = \texttt{str}(\tilde{t}_i)\}$
        **if** $|\texttt{valid\_splits}| = 0$ **then**
            **break**
        **else**
            $(i, t_1', t_2') \leftarrow \text{Random}(\texttt{valid\_splits})$
        **end if**
        $\tilde{\mathbf{t}} \leftarrow \left(\tilde{\mathbf{t}}_{<i}, t_1', t_2', \tilde{\mathbf{t}}_{>i}\right)$
    **end for**
    **Return** $\tilde{\mathbf{t}}$

---

# B. Additional Experimental Details

Here, we provide additional details on the experimental setup, including the hardware used, the dataset and models used, as well as details on the generation process. The code for our experiments is available at `https://github.com/Human-Centric-Machine-Learning/token-pricing`.

**Hardware setup.** Our experiments are executed on a compute server equipped with $2 \times$ Intel Xeon Gold 5317 CPU, 1,024 GB main memory, and $2 \times$ H100 Nvidia GPU (80 GB, Hopper Architecture). In each experiment, a single Nvidia H100 GPU is used.

**Datasets.** For the results presented across all figures, we generated model responses to prompts obtained from the LMSYS-Chat-1M dataset (Zheng et al., 2024). We use the LMSYS-Chat-1M dataset exclusively to obtain a varied sample of potential user prompts. We filter user prompts to obtain the first 1200 questions that are in English, Spanish, Russian and Chinese languages (by using the `language` keyword) and whose length (in number of characters) is in the range $[20, 100]$, to avoid trivial or overly elaborated prompts. Non-English prompts are only used in the results of Figure 11. We have repeated our experiments with a different set of 1200 randomly selected prompts from the LMSYS-Chat-1M dataset and have found indistinguishable results.

**Models.** In our experiments, we use the models `Llama-3.2-1B-Instruct` and `Llama-3.2-3B-Instruct` from the `Llama` family, the models `Gemma-3-1B-It` and `Gemma-3-4B-It` from the `Gemma` family, and `Ministral-8B-Instruct-2410`. The models are obtained from publicly available repositories from `Hugging Face`[8].

**Generation details.** For all experiments involving the LMSYS dataset, we use the `transformers` library in `Python` 3.11 to generate outputs of varying length between 200 and 300 tokens under various temperature and $p$ values. We use the system prompt "`You are a helpful assistant.  Be clear and concise.`", and compute standard deviations by running 5 independent generations with different seeds, and 90% symmetric confidence intervals for the mean values assuming a $t-$distribution value of 2.015. The 90% confidence intervals are shown in the plots.

For the experiments involving GPU energy measurements (Figure 10 and Table 1), we use the `pynvml` library in `Python` 3.11 to record the power consumption of the GPU during the generation and verification of a token sequence. During the experiments, we isolated the GPU so that the unique (energy-intensive) process running was the model execution.

**Licenses.** The LMSYS-Chat-1M dataset is licensed under the LMSYS-Chat-1M Dataset License Agreement.[9] The `Llama-3.2-1B-Instruct` and `Llama-3.2-3B-Instruct` models are licensed under the LLAMA 3.2 COMMUNITY LICENSE AGREEMENT.[10] The `Gemma-3-1B-It` and `Gemma-3-4B-It` models are licensed under the GEMMA TERMS OF USE.[11] The `Ministral-8B-Instruct-2410` model is licensed under the MISTRAL AI RESEARCH LICENSE.[12]

---

[8] `https://huggingface.co/meta-llama/Llama-3.2-3B-Instruct`
`https://huggingface.co/meta-llama/Llama-3.2-1B-Instruct`
`https://huggingface.co/google/gemma-3-1b-it`
`https://huggingface.co/google/gemma-3-4b-it`
`https://huggingface.co/mistralai/Ministral-8B-Instruct-2410`
[9] `https://huggingface.co/datasets/lmsys/lmsys-chat-1m`
[10] `https://ai.google.dev/gemma/terms`
[11] `https://www.gemma.com/gemma3_0/license/`
[12] `https://mistral.ai/static/licenses/MRL-0.1.md`

# C. Proofs

## C.1. Proof of Theorem 3.2

We prove the theorem by reduction from the Hamiltonian path problem (Karp, 2010), which is known to be NP-complete, to the problem of finding a plausible tokenization under top-$p$ sampling longer than a given number of tokens. Consequently, this will prove the hardness of the problem of finding a longest plausible token sequence $\tilde{\mathbf{t}}$ under top-$p$ sampling, as stated in Eq. 3. In the Hamiltonian path problem, we are given a directed graph $\mathcal{G}$, that is, a set of nodes $\mathcal{N} = \{1, \dots, n\}$ and a set of edges $\mathcal{E}$ between them, where $e = (\nu, \nu')$ denotes an edge from node $\nu$ to node $\nu'$. The goal is to decide whether there exists a path that visits all nodes exactly once.

The core idea of the construction is to represent a path in the graph $\mathcal{G}$ as a sequence of tokens, where each node $j \in \mathcal{N}$ is represented by a token consisting of $j$ times the character "a". In addition, we set the parameter $p \in (0, 1)$ of top-$p$ sampling and the next-token distributions of the LLM such that a token sequence $\tilde{\mathbf{t}}$ with $\mathrm{str}\,(\tilde{\mathbf{t}}) = \mathrm{str}(\mathbf{t})$ and $\mathrm{len}\,(\tilde{\mathbf{t}}) > 1$ is plausible if and only if the tokens in $\tilde{\mathbf{t}}$ correspond to a Hamiltonian path in the graph $\mathcal{G}$.

We proceed with the construction as follows. Let $\Sigma = \{\text{"a"}\}$ be the alphabet and the LLM's vocabulary be

$$\mathcal{V} = \{\text{"a"}, \text{"aa"}, \dots, \underbrace{\text{"a...a"}}_{n \text{ times}}, \underbrace{\text{"a...a"}}_{\lambda \text{ times}}, \varnothing\},$$

where $\lambda = \sum_{j=1}^{n} j = n(n+1)/2$ and $\varnothing$ denotes the end-of-sequence token. Moreover, let the true output token sequence $\mathbf{t}$ consist of a single token—the one that contains $\lambda$ times the character "a". Further, to keep the notation concise, we refer to the set of the first $n$ tokens in $\mathcal{V}$ as $\mathcal{V}_n$. Then, we define a mapping $\Phi \colon \mathcal{V}_n \to \mathcal{N}$ from tokens to nodes as

$$\Phi(\underbrace{\text{"a...a"}}_{j \text{ times}}) = j \text{ for } j = 1, \dots, n.$$

We fix the parameter $p$ and a next-token distribution of the LLM such that, given a (partial) token sequence $\tilde{\mathbf{t}} = (\tilde{t}_1, \dots, \tilde{t}_k)$, the restricted set of tokens $\mathcal{V}_p\,(\tilde{\mathbf{t}})$ from which the LLM can sample the next token is given by

$$\mathcal{V}_p\,(\tilde{\mathbf{t}}) = \begin{cases} \{\varnothing\} & \text{if } \left|\mathrm{str}\,(\tilde{\mathbf{t}})\right| \geq \lambda \\ \mathcal{V} \setminus \varnothing & \text{if } \tilde{\mathbf{t}} = () \\ \{v \in \mathcal{V}_n : v \neq \tilde{t}_i \text{ for all } i \in [k] \text{ and } \left(\Phi\,(\tilde{t}_k), \Phi\,(v)\right) \in \mathcal{E}\} \cup \{\varnothing\} & \text{otherwise.} \end{cases} \quad (6)$$

In words, the last case states that the LLM can sample any token consisting of up to $n$ times the character "a" as long as it is not already in the sequence $\tilde{\mathbf{t}}$, that is, the corresponding node has not been visited yet, and there is an edge in the graph $\mathcal{G}$ connecting that node to the node corresponding to the last token in $\tilde{\mathbf{t}}$. When the sequence $\tilde{\mathbf{t}}$ is empty (*i.e.*, the path has not started yet), the LLM can sample any token in $\mathcal{V}$ except for the end-of-sequence token $\varnothing$, which it is only allowed to sample when the sequence $\tilde{\mathbf{t}}$ contains at least $\lambda$ characters.

We can now show that a Hamiltonian path in the graph $\mathcal{G}$ exists if and only if the solution $\tilde{\mathbf{t}}$ to the optimization problem given by Eq. 3 has $\mathrm{len}\,(\tilde{\mathbf{t}}) > 1$.[13] Assume that the optimal solution to the problem is such that $\mathrm{len}\,(\tilde{\mathbf{t}}) > 1$. Then, $\tilde{\mathbf{t}}$ cannot contain the token that consists of $\lambda$ times the character "a" because this would imply that it consists of strictly more than $\lambda$ characters and, therefore, $\mathrm{str}(\tilde{\mathbf{t}}) \neq \mathrm{str}(\mathbf{t})$. Additionally, $\tilde{\mathbf{t}}$ cannot contain any token twice, as that would violate its plausibility according to Eq. 6. Therefore, it has to hold that $\tilde{\mathbf{t}}$ contains all tokens in $\mathcal{V}_n$ exactly once, since this is the only way to form a sequence that contains $\lambda = \sum_{j=1}^{n} j$ characters. This implies that there exists a sequence of edges $\left(\Phi\,(\tilde{t}_1), \Phi\,(\tilde{t}_2)\right), \dots, \left(\Phi\,(\tilde{t}_{n-1}), \Phi\,(\tilde{t}_n)\right)$ in the graph $\mathcal{G}$ that visits all nodes exactly once. Hence, a Hamiltonian path exists.

Now, assume that there exists a Hamiltonian path in the graph $\mathcal{G}$ that visits all nodes once, forming a sequence $(\nu_1, \nu_2, \dots, \nu_n)$ with $\nu_i \in \mathcal{N}$ and $\nu_i \neq \nu_j$ for $i \neq j$. Then, the corresponding token sequence $\mathbf{t}' = (t'_1, t'_2, \dots, t'_n)$ with $\Phi\,(t'_i) = \nu_i$ for $i \in [n]$ is a valid tokenization of the string $\mathrm{str}(\mathbf{t})$ since $\sum_{i=1}^{n} |\mathrm{str}(t'_i)| = \sum_{i=1}^{n} \nu_i = \lambda$. Moreover, the sequence $\mathbf{t}'$ is plausible by construction and satisfies $\mathrm{len}\,(\mathbf{t}') = n > 1 = \mathrm{len}\,(\tilde{\mathbf{t}})$. Finally, note that if $\mathcal{G}$ does not admit a Hamiltonian

---

[13]For ease of exposition, we assume that the end-of-sequence token $\varnothing$ does not contribute to the length of the sequence $\tilde{\mathbf{t}}$.

path, then `str(t)` cannot be tokenized as a sequence of plausible tokens in $\mathcal{V}_n$. Hence, the only plausible tokenization is the token with $\lambda$ characters, which has length 1. This concludes the proof.

In what follows, we present two extensions of the reduction to other settings where a provider may want to misreport the output token sequence without raising suspicion. Specifically, we consider the case where the provider reports a token sequence $\tilde{t}$ that is plausible under top-$k$ sampling and the case where the provider reports a token sequence $\tilde{t}$ whose probability is greater than a given threshold.

### C.1.1. Hardness of Finding the Longest Plausible Tokenization under Top-$k$ Sampling

Top-$k$ sampling is an approach of filtering out low-probability tokens during the sampling process, similar to top-$p$ sampling. In top-$k$ sampling, given a partial token sequence $\tilde{t}$, the LLM samples the next token from the set of $k$ most probable tokens $\mathcal{V}_k\left(\tilde{t}\right)$ at each step of the autoregressive process, where $k \in \{1, \ldots, |\mathcal{V}| - 1\}$. In this setting, the problem of finding the longest tokenization of a given output token sequence $t$ that is plausible under top-$k$ sampling is NP-Hard with the core idea of the reduction being similar to the one for top-$p$ sampling.

The main difference lies in the fact that, in top-$k$ sampling, the restricted set of tokens $\mathcal{V}_k\left(\tilde{t}\right)$ needs to have a fixed size $k$ in contrast to the construction of $\mathcal{V}_p\left(\tilde{t}\right)$ in Eq. 6, which is a variable size set. To ensure that similar arguments for establishing a one-to-one correspondence between a Hamiltonian path in the graph $\mathcal{G}$ and a plausible token sequence $\tilde{t}$ of length greater than 1 still hold, one can construct the set $\mathcal{V}_k\left(\tilde{t}\right)$ using a similar approach as in Eq. 6 but also including "padding" tokens that do not correspond to any node in the graph $\mathcal{G}$ to maintain a fixed size. To this end, we can maintain the same true output token sequence $t$, consisting of $n(n+1)/2$ times "a" and augment the vocabulary $\mathcal{V}$ of the previous construction by adding $n$ additional tokens

$$\mathcal{V}_b = \{\text{"b", "bb"}, \ldots, \underbrace{\text{"b...b"}}_{n \text{ times}}\}$$

that are irrelevant for the string $s = \text{str}(t)$, do not correspond to any node in the graph $\mathcal{G}$, and do not affect the mapping $\Phi$.

Then, note that, the set $\mathcal{V}_p\left(\tilde{t}\right)$ in Eq. 6 contains at most $n + 1$ tokens. Here, the idea is to set $k = n + 1$ and to construct the set $\mathcal{V}_k\left(\tilde{t}\right)$ as follows:

$$\mathcal{V}_k\left(\tilde{t}\right) = \mathcal{V}_p\left(\tilde{t}\right) \cup G\left(\mathcal{V}_p\left(\tilde{t}\right)\right), \tag{7}$$

where $G\left(\mathcal{V}_p\left(\tilde{t}\right)\right)$ is the set of the first $n + 1 - |\mathcal{V}_p\left(\tilde{t}\right)|$ tokens in $\mathcal{V}_b$. Since the additional tokens in $G\left(\mathcal{V}_p\left(\tilde{t}\right)\right)$ are not part of the mapping $\Phi$ and cannot be used to tokenize the string $s = \text{str}(t)$, they influence neither the plausibility of the optimal solution to the problem of Eq. 3 nor the corresponding Hamiltonian path in the graph $\mathcal{G}$. Therefore, the same arguments as in the proof of Theorem 3.2 hold, and we conclude that the problem of finding a longest tokenization of a given output token sequence $t$ that is plausible under top-$k$ sampling is NP-Hard.

### C.1.2. Hardness of Finding the Longest Tokenization Whose Generation Probability Is Greater than a Threshold

We now focus on a slightly different setting where the provider reports a token sequence $\tilde{t}$ under the plausibility condition that the LLM does not assign a very low probability to the sequence as a whole. Formally, we require that the probability of the LLM generating the token sequence $\tilde{t}$ satisfies

$$P\left(\tilde{t}\right) := P\left(\tilde{t}_1\right) \prod_{i=2}^{k} P\left(\tilde{t}_i \mid \tilde{t}_{<i}\right) \geq \varepsilon, \tag{8}$$

where $\varepsilon$ is a user-specified threshold and $P\left(\tilde{t}_i \mid \tilde{t}_{<i}\right)$ is the probability of the LLM generating the token $\tilde{t}_i$ given the previously generated tokens $\tilde{t}_{<i} = \left(\tilde{t}_1, \ldots, \tilde{t}_{i-1}\right)$. In this setting, the problem of finding a longest tokenization under Eq. 8 is also NP-hard. Similar as before, the proof is to set the next-token distributions of the LLM in a way that assigns low probability to token sequences that do not lead to a Hamiltonian path in $\mathcal{G}$. Specifically, let $\delta$ be a constant such that $0 < \delta < 1/(n+1)$, and assume all next-token distributions are such that, given $\left(\tilde{t}_1, \ldots, \tilde{t}_k\right)$, assign probability mass $(1 - \delta)/n$ to each of the tokens in

$$\mathcal{H}_i := \left\{v \in \mathcal{V}_n : v \neq \tilde{t}_i \text{ for all } i \in [k] \text{ and } \left(\Phi\left(\tilde{t}_k\right), \Phi\left(v\right)\right) \in \mathcal{E}\right\}, \tag{9}$$

$\delta$ to each of the tokens in $\mathcal{V}_n \setminus \mathcal{H}_i$, 0 to the token with $\lambda$ times the character "a", and any remaining probability mass to the end-of-sequence token $\varnothing$.[14] The high-level idea here is to set the probabilities of next tokens in such a way that the LLM assigns very low probability to the entire token sequence $\tilde{\mathbf{t}}$ if it concatenates two tokens whose corresponding nodes are not connected via an edge in the graph $\mathcal{G}$ or if the latter token has already been used in the sequence.

Given this construction, we set the user-specified threshold as $\varepsilon = \left(\frac{1-\delta}{n}\right)^n$. Now, given a Hamiltonian path in the graph $\mathcal{G}$ that visits all nodes once and forms a sequence $(\nu_1, \nu_2, \ldots, \nu_n)$ with $\nu_i \in \mathcal{N}$ and $\nu_i \neq \nu_j$ for $i \neq j$, the corresponding token sequence $\mathbf{t}' = (t'_1, t'_2, \ldots, t'_n)$ has cumulative probability exactly $\varepsilon$, so it is plausible and has length greater than 1. Reciprocally, given a plausible tokenization $\tilde{\mathbf{t}}$ with length greater than 1, the corresponding sequence $\left(\Phi\left(\tilde{t}_1\right), \Phi\left(\tilde{t}_2\right)\right), \ldots, \left(\Phi\left(\tilde{t}_{n-1}\right), \Phi\left(\tilde{t}_n\right)\right)$ has to be a Hamiltonian path. If this is not true, at least one of the tokens in $\tilde{\mathbf{t}}$ does not belong in its respective set $\mathcal{H}_i$ defined by Eq. 9, and hence the probability of the sequence $\tilde{\mathbf{t}}$ is at most

$$P\left(\tilde{\mathbf{t}}\right) \leq \delta \left(\frac{1-\delta}{n}\right)^{n-1} < \varepsilon, \tag{10}$$

which contradicts the assumption that $\tilde{\mathbf{t}}$ is plausible.

### C.2. Proof of Proposition 4.2

We will prove the proposition by contradiction. Assume that, under an incentive-compatible pricing mechanism $r$, there exist token sequences $\tilde{\mathbf{t}}, \tilde{\mathbf{t}}' \in \mathcal{V}^*$ such that $\mathrm{str}(\tilde{\mathbf{t}}) = \mathrm{str}(\tilde{\mathbf{t}}')$ and $r(\tilde{\mathbf{t}}) \neq r(\tilde{\mathbf{t}}')$ where, without loss of generality, it holds that $r(\tilde{\mathbf{t}}') > r(\tilde{\mathbf{t}})$. Now, consider that the output token sequence generated by the LLM is $\tilde{\mathbf{t}}$ and the provider runs a policy $\pi$ that reports $\tilde{\mathbf{t}}' \sim \pi(\tilde{\mathbf{t}})$. Moreover, let the policy $\pi$ be such that the energy costs of GPU computations it requires are zero, i.e., $c_\pi(\tilde{\mathbf{t}}) = 0$.[15] Then, it follows that

$$r(\tilde{\mathbf{t}}') > r(\tilde{\mathbf{t}}) \overset{(*)}{\Longrightarrow} r(\tilde{\mathbf{t}}') - c_{\mathrm{gen}}(\tilde{\mathbf{t}}) - c_\pi(\tilde{\mathbf{t}}) > r(\tilde{\mathbf{t}}) - c_{\mathrm{gen}}(\tilde{\mathbf{t}}) - c_{\pi_0}(\tilde{\mathbf{t}}) \implies U_\pi(\tilde{\mathbf{t}}', \tilde{\mathbf{t}}) > U_{\pi_0}(\tilde{\mathbf{t}}, \tilde{\mathbf{t}}),$$

where $(*)$ holds because the truthful policy $\pi_0$ also requires no GPU computations, i.e., $c_{\pi_0}(\tilde{\mathbf{t}}) = 0$. The latter inequality contradicts the fact that the pricing mechanism $r$ is incentive-compatible and concludes the proof.

### C.3. Proof of Theorem 4.3

Let $\mathbf{t}' = (t'_1, \ldots, t'_k)$ be the tokenization of the string $s = \mathrm{str}(\mathbf{t})$ that consists only of single-character tokens, i.e., $\mathrm{str}(\mathbf{t}) = \mathrm{str}(\mathbf{t}')$ with $|\mathrm{str}(\mathbf{t}')| = |\mathrm{str}(\mathbf{t})| = k$. Note that such a tokenization exists, since $\Sigma \subseteq \mathcal{V}$. From Proposition 4.2, we get

$$r(\mathbf{t}) = r(\mathbf{t}') \overset{(*)}{=} \sum_{i=1}^{k} r(t'_i) = \sum_{i=1}^{k} \sum_{\sigma \in \Sigma} \mathbb{1}[t'_i = \sigma] \cdot r(\sigma)$$

$$= \sum_{\sigma \in \Sigma} \mathrm{count}_\sigma(\mathbf{t}') \cdot r(\sigma) \overset{(**)}{=} \sum_{\sigma \in \Sigma} \mathrm{count}_\sigma(\mathbf{t}) \cdot r(\sigma),$$

where $\mathbb{1}$ denotes the indicator function, $(*)$ holds because the pricing mechanism is additive and $(**)$ holds because $\mathrm{str}(\mathbf{t}') = \mathrm{str}(\mathbf{t})$.

---

[14]Using the assumption that $\delta < 1/(n+1)$, it is easy to verify that the above construction leads to a valid probability distribution.
[15]Note that, such policies exist, as exemplified by Algorithm 2.

# D. Additional Experimental Results

## D.1. Performance of Algorithm 1 under Different LLMs and Temperature Values

In this section, we evaluate Algorithm 1 across different LLMs and temperatures.

Figure 4 shows the fraction of generated outputs for which Algorithm 1 finds a longer plausible tokenization. We observe that the higher the values of $p$ and temperature, the higher the likelihood that Algorithm 1 finds plausible longer tokenizations.

Figure 5 shows the percentage of tokens overcharged by an unfaithful provider who uses Algorithm 1. We observe that the percentage of overcharged tokens is unimodal with respect to the number of iterations $m$, and the higher the value of $p$ and the temperature, the higher the percentage of overcharged tokens, as the top-$p$ sets become larger and the likelihood that a longer tokenization is plausible increases.

Figure 6 shows the relative increase in utility that a provider using Algorithm 1 can achieve under pay-per-token compared to the faithful reporting policy $\pi_0$, as a function of their margin $\rho$ for output tokens. We observe that providers with low margins can benefit the most from misreporting, since their utility when faithfully reporting is reduced.

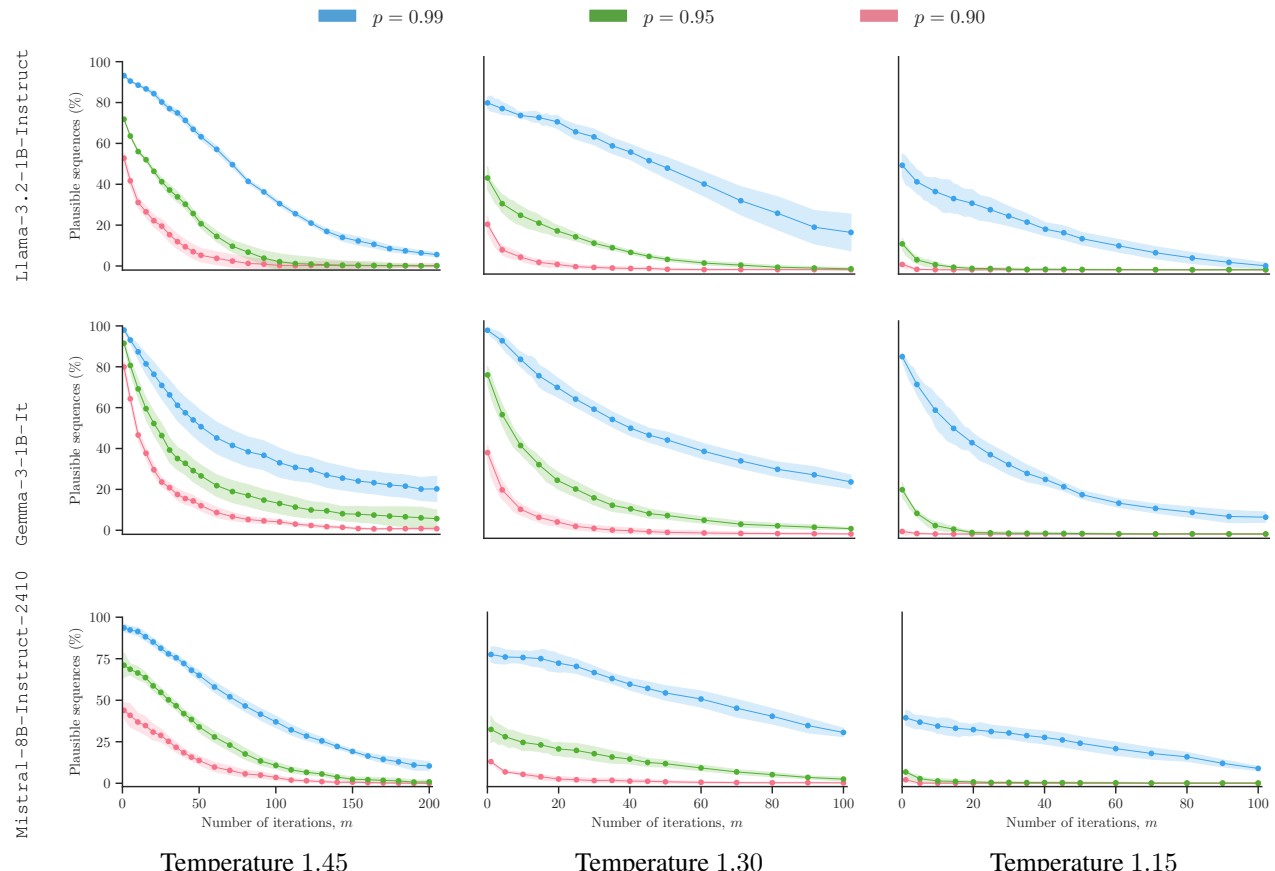

*Figure 4.* **Fraction of outputs for which Algorithm 1 finds a longer plausible tokenization.** The figure shows the fraction of outputs generated by different LLMs to 600 prompts from the LMSYS Chatbot Arena platform for which Algorithm 1 finds a longer plausible tokenization under top$-p$ sampling, for values of $m$, $p$ and temperature. We repeat each experiment 5 times to calculate 90% confidence intervals.

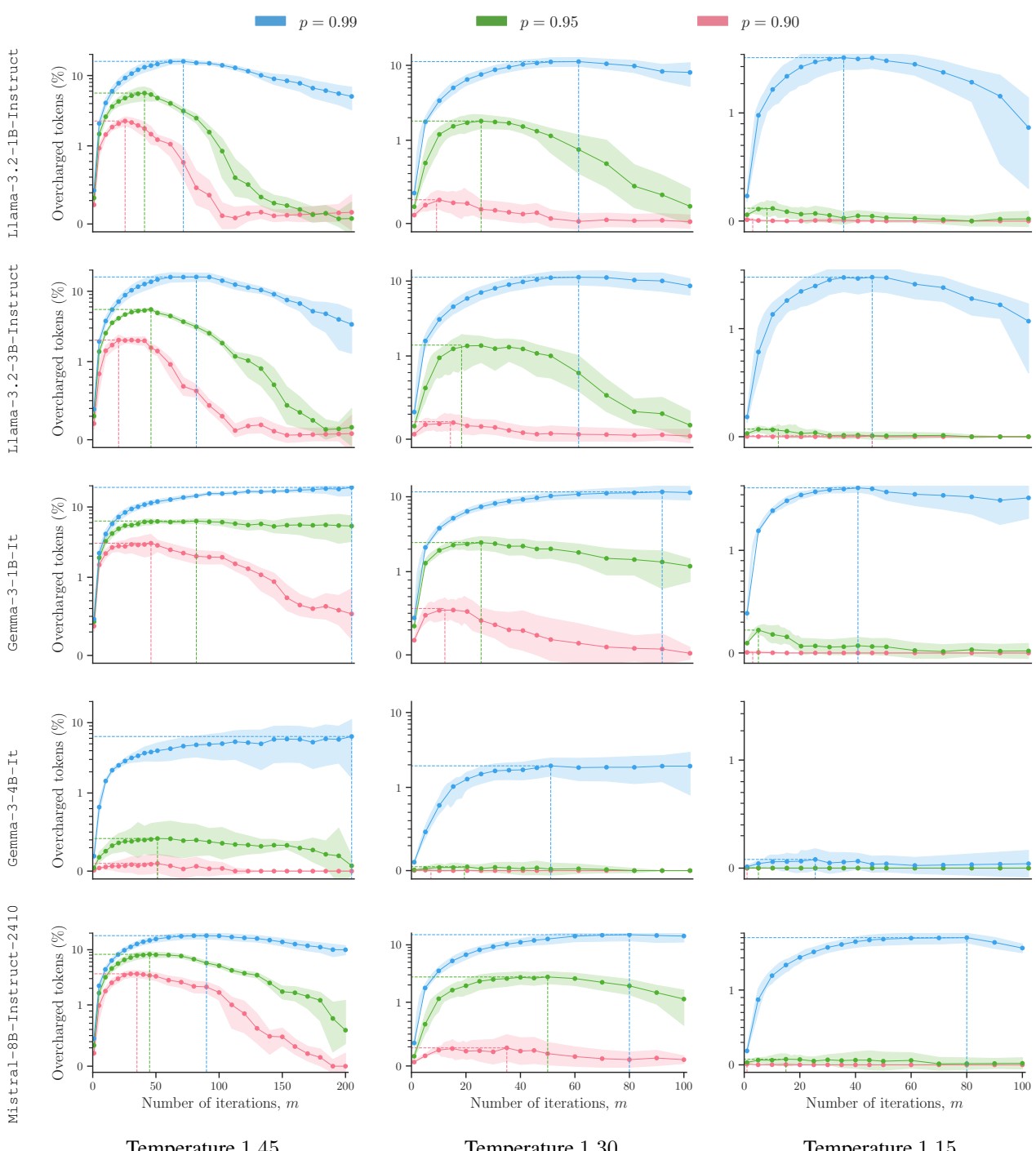

Temperature 1.45                 Temperature 1.30                 Temperature 1.15

*Figure 5.* **Additional revenue from misreporting the tokenization of outputs using Algorithm 1.** The panels show the percentage of tokens overcharged by an unfaithful provider who misreports the tokenization of the outputs generated by several LLMs to 600 prompts from the LMSYS Chatbot Arena platform using Algorithm 1, for different values of $m$, $p$ and temperature. The dashed lines highlight the optimal value of $m$. We repeat each experiment 5 times to obtain 90% confidence intervals.

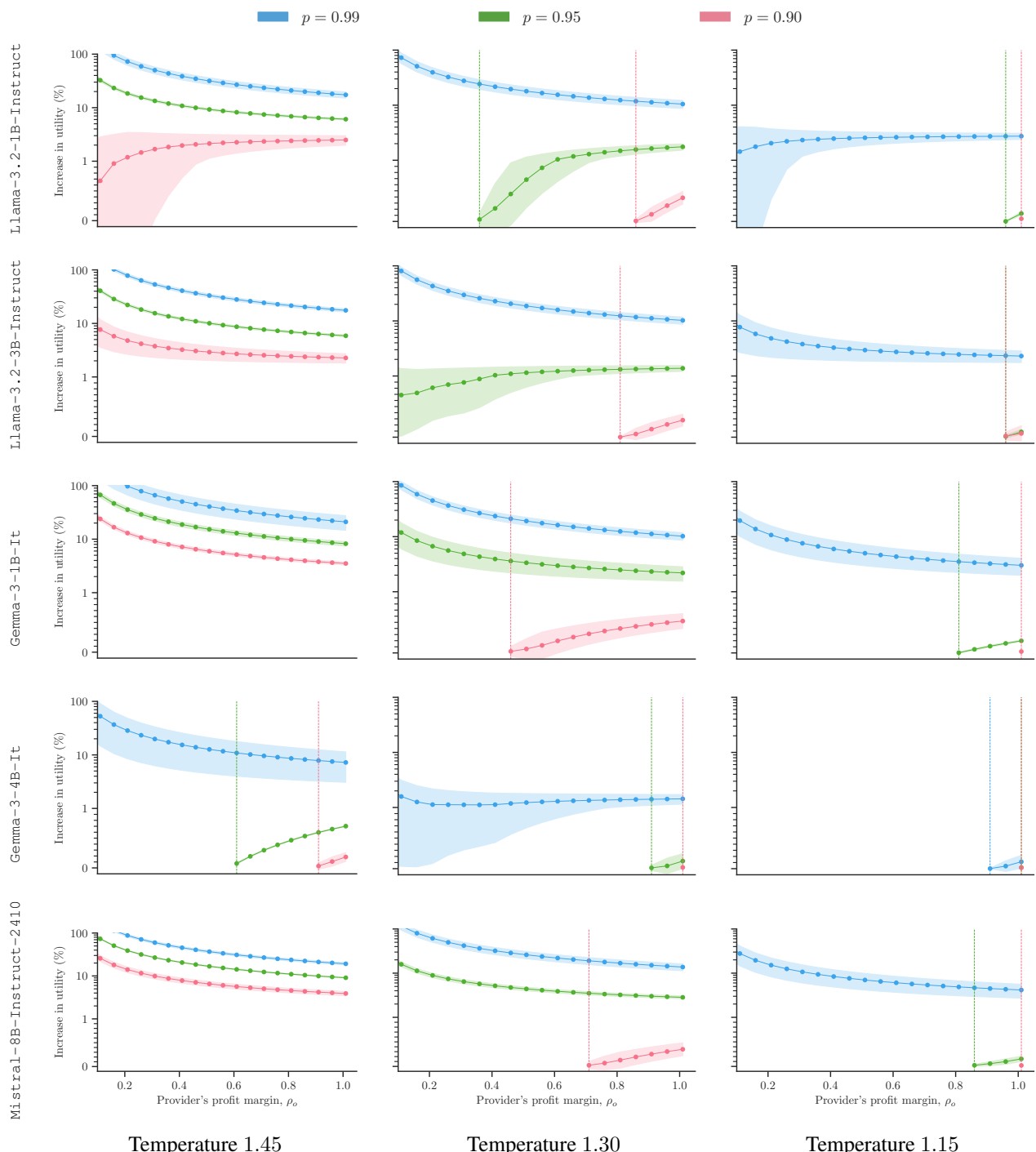

*Figure 6.* **Financial gain from misreporting the tokenization of outputs using Algorithm 1.** The panels show the financial gain that an unfaithful provider who misreports the tokenization of the outputs generated by several LLMs to 600 prompts from the LMSYS Chatbot Arena platform using Algorithm 1 can achieve, as a function of the profit margin on the output tokens $\rho_o$ and $p$. The dashed vertical lines represent the minimum margin under which misreporting is financially viable according to Eq. 4. Here, for each value of $p$, we select the optimal number of iterations $m$ in Algorithm 1 shown on Figure 2, and repeat each experiment 5 times to obtain 90% confidence intervals.

## D.2. Examples of Plausible Output Token Sequences Found by Algorithm 1

To illustrate how Algorithm 1 works, here, we provide examples of output token sequences generated by the `Llama-3.2-1B-Instruct` model, where the algorithm has found plausible tokenizations that are longer than the original output token sequence. Across all examples, we use the `Llama-3.2-1B-Instruct` model and set $p = 0.95$ and the temperature of the model to 1.3. We select prompts from the LMSYS dataset. For each example, we show (i) the true output token sequence generated by the model, and (ii) the modified output token sequence returned by Algorithm 1. We use "|" to indicate separations between tokens as generated by the model, and we use "|" to indicate the split points of the tokens that result from Algorithm 1. The number above each red separator indicates the iteration of the algorithm in which the respective token was split. We show all iterations until the sequence first becomes non-plausible.

```
...  The| third| film| appears| to| delve| into| the| themes| of| societal|
reaction| and|...  Here| are| movies| that| offer| similar| thematic
concerns|...
```

*(a)* True output token sequence

```
                                                        (1)
...  The| third| film| appears| to| del |ve| into| the| themes| of|
         (2)
soci |etal| reaction| and|...  Here| are| movies| that| offer| similar|
                    (3)
thematic conce | rns|...
```

*(b)* Modified output token sequence

*Figure 7.* Responses to the prompt "`is Dead Snow worth watching or should I watch directly Dead Snow 2?`".

```
...  Here| are| a| few| options| :|
1|.| **| T|rello|**:| T|rello| is| a| visual| project| management| tool|...
2|.| **| J|IRA|**:| As| mentioned|,| J|IRA| is| a| popular| At|lassian| suite|...
```

*(a)* True output token sequence

```
...  Here| are| a| few| options| :|
                        (1)
1|.| **| T|rello|** | :| T|rello| is| a| visual| project| management| tool|...
                 (2)                                              (3)
2|.| **| J|IRA|** | :| As| mentioned|,| J|IRA| is| a| popular| At|las | sian|
suite|...
```

*(b)* Modified output token sequence

*Figure 8.* Responses to the prompt "`What is a good tool to plan a complex server deployment?`".

```
The| easiest| way| to| invest| in| property|…  Real| estate| investment|
trusts| or| RE|IT|s|,| real| estate| mutual| funds| may| be| the| easiest|.|…
There| are| many| options| for| acquiring| income| such| as| ground| level|
rental| or| owning| a| building| through| a| partnership|.| The| highest|
performing| investment| may| remain| a| gamble| and| have| no| guarantee|.|
The| next| highest| would| have| to| be| investing| in| stocks| and| bonds|,
the| old| main|stay|.| Div|idend| and| bonds| have| higher| reliability|…
Note|:| the| previous| responses| and| answers| have| been| simplified|…
```

*(a)* True output token sequence

```
                (8)
The| eas | iest| way| to| invest| in| property|…  Real| estate| investment|
        (2)
trust | s| or| RE|IT|s|,| real| estate| mutual| funds| may| be| the| easiest|.|…
                                          (6)
There| are| many| options| for| acqu | iring| income| such| as| ground|
                    (7)
level| rental| or| ow | ning| a| building| through| a| partnership|.| The|
                                              (3)
highest| performing| investment| may| remain| a| gam | ble| and| have| no|
guarantee|.| The| next| highest| would| have| to| be| investing| in| stocks|
                        (4)                 (1)
and| bonds|, the| old| main|st | ay|.| Div|id | end| and| bonds| have| higher|
      (2)
reli | ability|…  Note|:| the| previous| responses| and| answers| have| been|
      (5)
simpl | ified|…
```

*(b)* Modified output token sequence

*Figure 9.* Responses to the prompt "What is currently the easiest investment opportunity with the capital and the highest game?".

### D.3. Energy cost of generation and verification of plausibility

In this section, we provide empirical measurements of the (GPU) energy consumption required to (i) generate an output sequence of tokens and (ii) compute the next-token probabilities for a given sequence of tokens. Here, note that (ii) is required to verify if such a sequence is plausible under top-$p$ sampling, as discussed in Section 3.2 and Section 3.3.

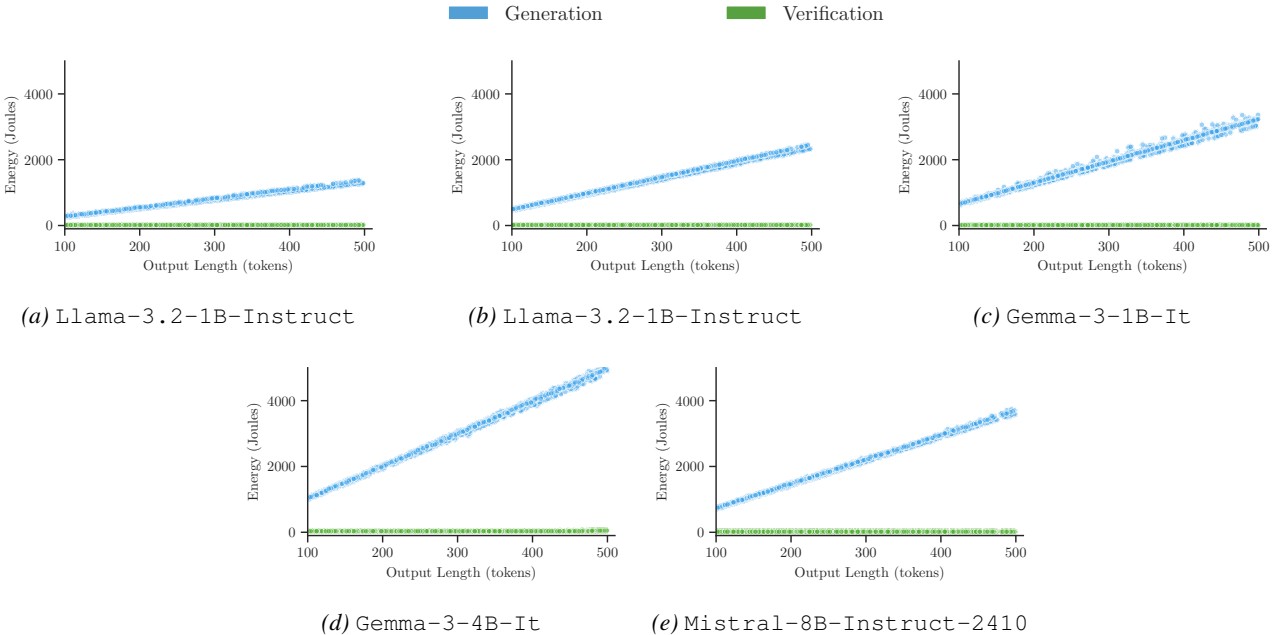

*(a)* `Llama-3.2-1B-Instruct`  *(b)* `Llama-3.2-1B-Instruct`  *(c)* `Gemma-3-1B-It`

*(d)* `Gemma-3-4B-It`  *(e)* `Mistral-8B-Instruct-2410`

*Figure 10.* **Energy cost of autoregressive generation vs. verification.** The figure shows, across different model families and for the first 1000 LMSYS prompts, the energy cost of autoregressively generating an output and the energy cost of verifying the same generated output (*i.e.*, computing the next-token probabilities). We measure energy cost as the power consumed by the GPU, a single Nvidia H100, integrated over the execution time, as calculated by Python's `pynvml` library. All executions are with temperature 1, no top-$p$ sampling, and with KV caching enabled.

*Table 1.* **Energy cost of autoregressive generation vs. verification.** The table shows, for different model families, the per-token energy cost $c_\circ$ for autoregressively generation, the per-output energy cost of verifying the same generated output $c_v$ (which, as shown in Figure 10, is approximately independent of sequence length), and the ratio between these two quantities. All values are computed over 400 prompts from the LMSYS Chatbot Arena platform. All executions are with temperature 1, no top-$p$ sampling, and with KV caching enabled on a single Nvidia H100. We repeat each experiment 5 times to obtain 90% confidence intervals.

| LLM | Per-token gen. cost $c_\circ$ (J) | Per-output verif. cost $c_v$ (J) | $c_\circ/c_v$ |
|---|---|---|---|
| `Llama-3.2-1B-Instruct` | $2.642 \pm 0.004$ | $15.188 \pm 0.005$ | $0.174 \pm 0.009$ |
| `Llama-3.2-3B-Instruct` | $4.805 \pm 0.004$ | $15.901 \pm 0.007$ | $0.302 \pm 0.009$ |
| `Gemma-3-1B-It` | $6.481 \pm 0.006$ | $12.730 \pm 0.002$ | $0.509 \pm 0.016$ |
| `Gemma-3-4B-It` | $9.993 \pm 0.005$ | $33.5 \pm 0.1$ | $0.298 \pm 0.005$ |
| `Ministral-8B-Instruct-2410` | $7.349 \pm 0.009$ | $17.743 \pm 0.008$ | $0.413 \pm 0.016$ |

### D.4. Provider's profit margin under pay-per-character for different languages

In Figure 11 we show, stratified by language, the distribution of a provider's margin across responses to prompts from the LMSYS Chatbot Arena platform after transitioning to a pay-per-character mechanism with character price $r_c = r_o \cdot \texttt{tpc}$, where $\texttt{tpc}$ is the (empirical) average token-to-character ratio computed over multilingual prompts.

We find that languages such as Russian or Chinese, which are likely underrepresented during tokenizer training in the models we study, result in lower (average) provider margins compared to their previous pay-per-token margin $\rho_o$, while better-represented languages like English or Spanish yield a slight margin increase. This occurs because, under pay-per-character pricing, the provider's margin for a given output decreases as the token-to-character ratio increases, and languages that differ significantly from English tend to produce outputs with higher token-to-character ratios (Petrov et al., 2023; Zhang et al., 2022). As a consequence, by reducing their output prices, pay-per-character partially alleviates the financial burden faced by users of minority languages (Ahia et al., 2023).

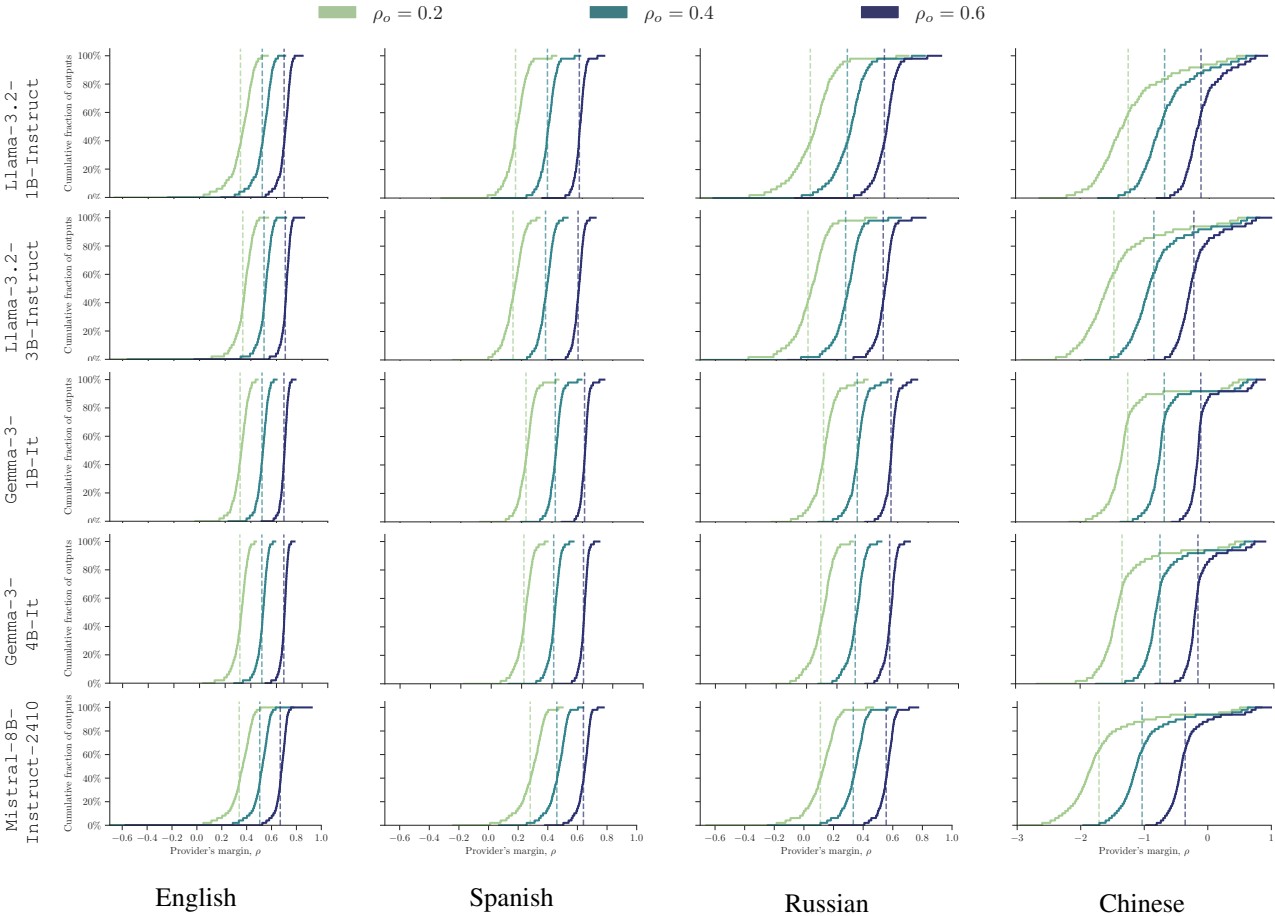

*Figure 11.* **Provider's profit margin under a pay-per-character pricing mechanism.** The panels show, across different languages, models and values $\rho_o$ of the provider's profit margin under pay-per-token, the (empirical) cumulative distribution function of their profit margin $\rho(\mathbf{t}) = 1 - c_{\text{gen}}(\mathbf{t})/r(\mathbf{t})$ per output $\mathbf{t}$ after their transition to pay-per-character. Here, we first compute the average ratio of number of tokens to number of characters ($\texttt{tpc}$) across the responses of each model to 600 multilingual prompts sampled from the LMSYS Chatbot Arena dataset. Then, we set $r_c = r_o \cdot \texttt{tpc}$ and compute the provider's profit margin $\rho(\mathbf{t})$ for outputs to 600 different monolingual prompts. In all panels, dashed vertical lines show empirical averages of the respective distributions, and we set the temperature of the models to 1.3 and use top-$p$ sampling with $p = 0.95$.

