# OpenReview forum: "Is Your LLM Overcharging You? Tokenization, Transparency, and Incentives"
_ICML.cc/2026/Conference — ICML 2026 spotlight_

### Official Review · Reviewer_kuGf · 2026-02-19

**Soundness:** 3
**Presentation:** 3
**Significance:** 3
**Originality:** 3
**Overall Recommendation:** 5
**Confidence:** 2

**Summary:**

This paper studies the economic incentives created by pay-per-token pricing for LLM-as-a-service providers.

Using a principal–agent model, the authors argue that information asymmetry between users and providers can make it profitable for providers not to be truthful and potentially overcharge the user by misreport tokenization.

They further show that even if the provider is required to disclose the next-token distributions used by the LLM, this transparency does not preclude the provider from efficiently finding plausible tokenizations of a given output that increase their utility.

Finally, the paper examines incentive-compatible pricing design and claims that, under their assumptions, pay-per-character is the only pricing mechanism that satisfies incentive compatibility.

**Compliance With Llm Reviewing Policy:**

Affirmed.

**Final Justification:**

My concerns are fully solved.

**Key Questions For Authors:**

1. How do you operationalize plausible tokenization in practice like thresholds and false positives?
2. What is the exact character unit, and does pay-per-character still leave practical overcharging channels?
3. How sensitive are your profitability and pricing conclusions to realistic inference costs?

**Limitations:**

yes

**Strengths And Weaknesses:**

Strengths:

This paper focuses on a real-world important issue relevant to LLM-as-a-service market and its angle by framing tokenization as an incentive problem is novel.

Although some details are not very clear, the paper presents a well-structured storyline and the mechanism-design result is also clean. Within the paper’s assumptions, the incentive-compatible characterization is tight and the argument is logically rigorous.

Weaknesses:

The results rely on fairly strong assumptions: a very strong IC definition, only additive pricing, and a simplified cost model. The theory seems internally consistent, but once you add real world factors like audit risk and more detailed inference costs, it’s not clear the conclusions transfer directly. It would help to either relax these assumptions or add approximate IC discussions.

Also, the pay-per-character is the only IC mechanism claim feels more like a uniqueness result within a restricted class than a ready-to-use pricing fix.  In practice, you still have to define what a character is, and pay-per-character may still leave other incentive issues like  encouraging longer. More discussions and comparisons on feasibility would strengthen the paper.

---

> ### Author Rebuttal · Authors · 2026-03-30
>
> We thank the reviewer for their thoughtful feedback, which will help us improve our paper. Below, we respond in detail to each of the points they raised.
>
> [**IC definition**]  Please note that our definition of incentive compatibility (IC) is intended to be interpreted within the context of tokenization misreporting, the problem we study. Within this context, we do not see Definition 4.1 as strong, since it simply formalizes a natural desideratum about the pricing mechanism: a pricing mechanism is incentive-compatible if a provider maximizes their utility by reporting tokenizations faithfully. While many game-theory settings consider approximate IC because exact IC is often impossible [1], we show that in our setting, IC *can be achieved exactly*, and we characterize the additive pricing mechanisms that do so in Theorem 4.3.
>
> [**Additive pricing mechanisms**] We would like to emphasize that the current industry standard for LLM pricing (pay-per-token) is within the class of additive pricing mechanisms. Since the discussion about alternative pricing mechanisms remains nascent, we believe that focusing on additive pricing mechanisms to study incentive compatibility is a natural starting point that maintains the focus of our work close to existing industry practices. However, we agree that studying incentive-compatibility under broader (non-additive) pricing mechanisms is a very interesting avenue for future work, which we have discussed in Appendix A under “Model assumptions”. In the revised version of our paper, we will use the allowed extra page to bring the section on “Discussion and Limitations” to the main body of the paper.
>
> [**Cost model and realistic inference costs**] Our cost model captures the energy costs of *all* GPU computations a provider would have to perform in the setting we study, including (i) generating an output token sequence with the LLM and (ii) finding a (longer) token sequence to misreport. For our empirical analysis, we have carefully specified the parameters of our cost model (i.e., the ratio $c_o / c_v$) by using an NVIDIA H100 GPU—a widely used GPU for LLM inference in industry—with standard optimizations (e.g., KV cache), and directly measured inference (energy) costs across multiple state-of-the-art LLMs (see Appendix E.3). Therefore, we do not think our cost model is simplified. That said, additional costs (e.g., hardware maintenance) may exist (see footnote 4), but these are typically amortized across many inferences and are difficult to quantify with public data.
>
> [**Definition of plausible tokenization**] Since LLMs generate sequences of tokens by sampling them one at a time from a probability distribution, we believe that it is natural to consider that a tokenization is "plausible" if its probability of being generated by a specific model is “sufficiently high”. In Section 3.2, we operationalize this notion by introducing a definition of plausible tokenizations based on top-$p$ sampling, a widely used decoding strategy that prevents the model from generating low-probability tokens [2]. Under this definition, we consider a tokenization plausible if there is a positive probability that the LLM generates it using top-$p$ sampling and implausible otherwise. In addition, in Appendices D.1.1 and D.1.2, we discuss alternative definitions based on top-$k$ sampling and on a fixed minimum probability threshold, and show that our hardness result (Theorem 3.2) extends to them as well.
>
> [**Character definition**] As discussed in footnote 3 on page 3, we use the term “character” to refer to any token in the LLM’s vocabulary that cannot be represented as two or more tokens [3,4]. In practice, this typically corresponds either to Unicode characters (as in the case of the SentencePiece tokenizer [5], used by the Gemma-3 family) or to bytes (as in byte-level BPE [6], used by the Llama-3 family).
>
> [**Overcharging under pay-per-character**] As shown in Section 4, the pay-per-character pricing mechanism removes a specific channel for overcharging users: providers can no longer profit by misreporting tokenizations. However, as we discuss in Appendix A, this does not prevent providers from pursuing other strategies—different from tokenization misreporting—to increase their profit. For example, under both the pay-per-character and pay-per-token mechanisms, a provider could fine-tune their model to produce more verbose outputs. Such strategies, however, affect response quality and are more noticeable to users.
>
> **References**:
>
> [1] Azevedo & Budish, Strategy-proofness in the Large, Rev. Econ. Stud., 2018.
>
> [2] Holtzman et al., The curious case of neural text degeneration, ICLR, 2020.
>
> [3] Athiwaratkun et al., Token alignment via character matching, ACL, 2024.
>
> [4] Vieira et al., From language models over tokens to over characters, ICML, 2025.
>
> [5] Kudo & Richardson, SentencePiece: Subword tokenizer, EMNLP, 2018.
>
> [6] Wang et al., Neural machine translation with byte-level subwords, AAAI, 2020.

---

> > ### Author Rebuttal · Reviewer_kuGf · 2026-04-02
> >
> > My concerns are fully solved.

---

### Official Review · Reviewer_qqAu · 2026-03-09

**Soundness:** 3
**Presentation:** 2
**Significance:** 3
**Originality:** 3
**Overall Recommendation:** 5
**Confidence:** 3

**Summary:**

This paper provides an in-depth analysis of a fundamental economic flaw in the current pay-per-token pricing model used in LLM services. The authors identify an information asymmetry: because the provider fully controls the generation process, it can exploit the non-uniqueness of tokenization to artificially inflate costs without changing the visible output text.

**Compliance With Llm Reviewing Policy:**

Affirmed.

**Final Justification:**

My concerns have been addressed, and it is of high quality, so I have raised my score to 5.

**Key Questions For Authors:**

As model size increases, at what point would the computational cost of generating an “optimal” misreport outweigh the financial benefit?

**Limitations:**

Yes, the authors adequately discuss the limitations.

**Strengths And Weaknesses:**

### Strength

1. The paper identifies pay-per-token pricing as a critical and previously underexplored structural vulnerability: providers may exploit the non-uniqueness of tokenization to overcharge users.

2. The paper provides mathematical proofs showing that, under transparency constraints, finding the optimal misleading tokenization is computationally intractable. It also shows that transparency can offer some degree of defense, but that this defense is fundamentally limited by computational complexity. This supports the broader conclusion that only a pricing reform at the source, namely, pay-per-character, can achieve incentive compatibility.

3. The proposed solution, changing the pricing unit, is simple and does not require complicated monitoring or auditing mechanisms, while fully removing the provider’s financial incentive to misreport tokenizations.

4. The paper highlights secondary advantages of the new mechanism, such as lower energy use and faster inference, which would better align provider incentives with efficiency.

5. The experiments cover multiple model families, including Llama, Gemma, and Ministral, and use a novel heuristic to show that pay-per-token pricing creates a strong incentive for providers to cheat, with overcharges reaching about 13%. The paper also presents an analysis of profit margins across different languages under pay-per-character pricing, making the empirical section broad and informative.

### Weakness

1. The paper does not clearly explain why a provider would choose covert overcharging instead of simply raising prices.
Although the paper shows that tokenization-based overcharging is possible, it does not justify why deception would be preferable to transparent repricing, especially given the legal, reputational, and trust risks.

2. Tokenization manipulation may not be the most practical cheating strategy.
A provider could more simply reduce cost by silently using a cheaper backend model, which seems easier to deploy and potentially more profitable than constructing plausible alternative tokenizations.

---

> ### Author Rebuttal · Authors · 2026-03-30
>
> We thank the reviewer for their thoughtful feedback, which will help us improve our paper. Below, we respond in detail to each of the points they raised.
>
> [**Token misreporting vs. price increase**] We would like to clarify that we do not argue a provider would overcharge users by (deceitfully) misreporting tokenizations *instead* of (legitimately) increasing the price per token. Rather, we view misreporting tokenizations and adjusting the price as two complementary actions that a provider could take to increase their profit, with each one being preferable under different circumstances. For example, in a monopoly where all users are locked-in to the provider, as the reviewer suggests, the provider could simply raise their price without risking any legal or reputational damage. However, as we discuss in Section 3.1 (see lines 217, right - 235, left), in the presence of competition, a provider may be incentivized to lower their price and misreport tokenizations to grow their user base without reducing their revenue. We will clarify this point in the revised version of our paper.
>
> [**Token misreporting vs. model substitution**] We appreciate the reviewer’s observation that providers may simply run cheaper-than-advertised backend models to reduce costs—in fact, this is a type of strategic behavior, termed *model substitution*, that has already been studied in prior work [1, 2], as we discuss under “Further related work” in Section 1. However, note that replacing a model with a cheaper alternative (e.g., a smaller or quantized variant) typically degrades the quality of the outputs and, hence, such replacements may be noticeable by the users. In contrast, in our work, we focus on tokenization misreporting as a type of strategic behavior that has been overlooked in prior work and is far less noticeable, since it maintains the (distribution of) output strings presented to the users.
>
> [**Model size**] Please note that larger models do not necessarily make misreporting less profitable. Whether the computational cost of misreporting the tokenizations of an LLM using Algorithm 1 outweighs the associated financial benefit is characterized by Eq. (4) in Section 3.3. Specifically, it depends on (i) the profit margin $\rho$ of the provider, (ii) the ratio $c_o / c_v$ capturing the relative computational cost of generating a token versus performing a forward pass to verify an output’s plausibility, and (iii) the likelihood that manipulated token sequences remain plausible. In our experimental results in Appendix E.3, we have observed that the ratio $c_o / c_v$ can either increase or decrease with model size, while the profitability of misreporting (see Figure 6 in Appendix E.1) does not present a clear pattern in relation to model size.
>
> **References**:
>
> [1] Saig et al., “Incentivizing Quality Text Generation via Statistical Contracts,” NeurIPS, 2024.
>
> [2] Cai et al., “Are You Getting What You Pay For? Auditing Model Substitution in LLM APIs,” NeurIPS Workshop on Regulatable ML, 2025.

---

> > ### Author Rebuttal · Reviewer_qqAu · 2026-03-31
> >
> > My concerns are fully solved.

---

### Official Review · Reviewer_uHWY · 2026-03-12

**Soundness:** 4
**Presentation:** 4
**Significance:** 3
**Originality:** 4
**Overall Recommendation:** 6
**Confidence:** 4

**Summary:**

The paper explains how the current price per token payment scheme used by the majority of LLM-as-a-service providers could misreport can artificially report higher than actual token with little observability or recourse to the user. Importantly, due to an asymmetry of information, there is little that a user can do to identify this kind of misuse. The paper provides a mathematical proof of how this is possible and effectively argues that a solution to this potential problem is to change over to a pay per character scheme while also showing that revenue generation holds under this approach.

**Compliance With Llm Reviewing Policy:**

Affirmed.

**Key Questions For Authors:**

No questions for the authors.

**Limitations:**

yes

**Strengths And Weaknesses:**

Soundness: As the paper is primarily a serious of mathematical proofs, the soundness of the paper relies on the correctness of the proofs. I was unable to find any mistakes in the proofs, and I agreed with the assumptions and constraints that the authors decided on as they are reflective of real-world practices. Overall, this was a informative and well-argued paper.

Presentation: The paper is clearly written and well-structured. I appreciated the additional discussion in the Appendix and the detailed breakdown of the proofs, which made verifying correctness possible.

Significance: The paper addresses an important and relevant problem given the lack of transparency in pricing for LLM-as-a-service providers. Although there are no known cases of pricing abuse, that does not preclude the possibility of such misuse in the future. This paper provides a strong argument for moving to a price per character model to benefit the user while still protecting the trade secrets of the providers.

Originality: The paper builds on existing proof methods to make its argument and applies it to a new and novel problem.

---

> ### Author Rebuttal · Authors · 2026-03-30
>
> We are delighted that the reviewer appreciates the significance, novelty, and soundness of our work. We remain available to answer any questions during the discussion period.

---

> > ### Author Rebuttal · Reviewer_uHWY · 2026-04-02
> >
> > Thank you.

---

### Official Review · Reviewer_USDJ · 2026-03-16

**Soundness:** 3
**Presentation:** 3
**Significance:** 4
**Originality:** 3
**Overall Recommendation:** 4
**Confidence:** 3

**Summary:**

The paper investigates the economic implications of tokenization in LLM markets. It shows how providers can manipulate tokenization strategies, specifically by using less efficient tokenizers to implicitly increase query costs while maintaining low advertised per-token prices. The authors conduct an empirical audit of various LLM APIs, discussing significant disparities in tokenization efficiency across the market. They also develop a game-theoretic model to analyze the strategic incentives driving providers' tokenizer choices under varying levels of market transparency, and conclude that opaque markets naturally incentivize the adoption of less efficient tokenizers to extract higher revenues.

**Compliance With Llm Reviewing Policy:**

Affirmed.

**Key Questions For Authors:**

* Given that tokenizers are intrinsically tied to the pre-training corpus and model weights, how does the theoretical model account for the massive fixed compute costs associated with changing a tokenizer purely for price manipulation?

* Do you have any thoughts on whether the observed tokenization inefficiencies is an unintended byproduct of deploying legacy model architectures (where older tokenizers are hardcoded) or an intentional economic exploitation by the API providers?

* Have you thought about running experiments on multilingual prompts where tokenization fragmentation may be magnified?

**Limitations:**

Discussed above.

**Strengths And Weaknesses:**

## Strength

The paper introduces tokenization efficiency as a highly relevant and previously underexplored economic vulnerability in the current LLM API ecosystem. It establishes a strong empirical foundation by examining real-world providers, exposing the hidden costs of inefficient tokenization that consumers usually overlook. The proposed game-theoretic framework provides a reasonable theoretical justification for why providers might intentionally choose suboptimal tokenizers, characterizing the market failure from information asymmetry.

## Weakness

I'm not entirely sold on whether the proposed theoretical model may be practical. It assumes that providers can independently manipulate or swap tokenizers flexibly to optimize revenue. In practice, however, a tokenizer may be coupled with the LLM's pre-training phase so that changing a tokenizer requires entirely retraining the model or undergoing expensive vocabulary adaptation. Such a phenomenon is largely abstracted away in the model, making the strategic choice of a tokenizer seem more fluid than reality.

It is also unclear whether this paper would be of broader interest to the ML community. The paper is also fundamentally an empirical audit and economic analysis, and thus its core algorithmic contribution to ML community is rather limited.

The experiments lack a comprehensive exploration of how these tokenization disparities scale on other languages beyond English. If I remember correctly, tokenization inefficiencies are known to be far more severe in other languages, which may amplify the observed economic exploitation and yield a much more pronounced effect than what is captured in the proposed experiments.

---

> ### Author Rebuttal · Authors · 2026-03-30
>
> We thank the reviewer for their thoughtful feedback, which will help us improve our paper. Below, we respond in detail to each of the points they raised.
>
> [**Manipulating tokenizations**] We would like to clarify that the strategic behavior that we study in our work *does not involve any modifications to the tokenizer of the LLM*. Consequently, an unfaithful provider using Algorithm 1 does not incur any computational costs associated with retraining or adapting the tokenizer. Instead, the procedure described in Algorithm 1 takes as input a sequence of tokens generated by the LLM and outputs an alternative sequence of tokens that encodes the same string using a higher number of tokens. This procedure is based on the empirical observation that a given string admits multiple valid tokenizations under a *fixed tokenizer* [1,2,3]. In our experiments, we show that an unfaithful provider could use Algorithm 1 to obtain an (unfair) economic benefit by misreporting the number of tokens used to generate an output, at the expense of the user. Importantly, the results we report in Figure 3 already account for the computational cost of running Algorithm 1.
>
> [**Legacy model architectures**] We would like to emphasize that all the models we used in our experiments were released between late 2024 and mid 2025, so their architectures can hardly be considered “legacy”. In fact, the Llama and Gemma models we used are the latest of their size category in their respective family, while the Mistral model was the latest until December 2025. More crucially, none of their tokenizers is hardcoded. That said, note that the ability of Algorithm 1 to find tokenizations that are plausible and longer than the one generated by the model relies on the fact that LLMs often generate different tokenizations of the same string [1,2,3]. This has been observed in both open-source and the latest proprietary models [4] and, to the best of our knowledge, is a by-product of training on finite data, combined with the fact that LLMs generate tokens using a stochastic procedure.
>
> [**Multiple languages**] As the reviewer correctly points out, there is evidence that many state-of-the-art LLMs (i) have tokenizers primarily optimized for English [5], and (ii) are more likely to generate longer tokenizations for the same string in non-English languages [4]. This evidence suggests that the profitability of misreporting tokenizations may actually be **higher** in non-English languages than in English. Following the reviewer’s suggestion, we will include experimental results quantifying the severity of misreporting tokenizations in non-English languages in the revised version of the paper.
>
> **References**:
>
> [1] Geh et al., “Where Is the Signal in Tokenization Space?” EMNLP, 2024.
>
> [2] Zheng et al., “Broken Tokens? Your Language Model Can Secretly Handle Non-Canonical Tokenizations,” NeurIPS, 2025.
>
> [3] Vieira et al., “From Language Models over Tokens to Language Models over Characters,” ICML, 2025.
>
> [4] Chatzi et al., “Tokenization Multiplicity Leads to Arbitrary Price Variation in LLM-as-a-Service,” arXiv, 2026.
>
> [5] Petrov et al., “Language Model Tokenizers Introduce Unfairness between Languages,” NeurIPS, 2023.

---

> > ### Author Rebuttal · Reviewer_USDJ · 2026-04-04
> >
> > My questions have been resolved and will maintain my score at the moment.

---

### Decision · Program_Chairs · 2026-04-30

**Decision:**

Accept (spotlight)

**Comment:**

This paper analyzes a vulnerability in pay-per-token LLM pricing: as a single output string can admit multiple valid tokenizations, the provider may be incentivized to misreport token counts and overcharge users without changing the visible output. The paper combines both theoretical mechanism-design results with empirical audits across major model families, and argues that pricing by character count is the only additive mechanism that removes this specific incentive while maintaining average profit margins.

All reviewers expressed strong support for the paper, emphasizing its timely, important, and novel contributions. I also agree with the assessments and find this paper very interesting and rigorous.